# KVzip: Query-Agnostic KV Cache Compression with Context Reconstruction

**Jang-Hyun Kim**[1 2], **Jinuk Kim**[1 2], **Sangwoo Kwon**[1], **Jae W. Lee**[1],
**Sangdoo Yun**[3], **Hyun Oh Song**[* 1 2]

[1]Seoul National University, [2]Neural Processing Research Center, [3]NAVER AI Lab

{blue378, hyunoh}@snu.ac.kr

https://github.com/snu-mllab/KVzip

## Abstract

Transformer-based large language models (LLMs) cache context as key-value (KV) pairs during inference. As context length grows, KV cache sizes expand, leading to substantial memory overhead and increased attention latency. This paper introduces *KVzip*, a query-agnostic KV cache eviction method enabling effective reuse of compressed KV caches across diverse queries. KVzip quantifies the importance of a KV pair using the underlying LLM to reconstruct original contexts from cached KV pairs, subsequently evicting pairs with lower importance. Extensive empirical evaluations demonstrate that KVzip reduces KV cache size by $3$-$4\times$ and FlashAttention decoding latency by approximately $2\times$, with negligible performance loss in question-answering, retrieval, reasoning, and code comprehension tasks. Evaluations include various models such as LLaMA3.1, Qwen2.5, and Gemma3, with context lengths reaching up to 170K tokens. KVzip significantly outperforms existing query-aware KV eviction methods, which suffer from performance degradation even at a 90% cache budget ratio under multi-query scenarios.

## 1 Introduction

Transformer-based LLMs with long-context capabilities have significantly enhanced real-world applications, including long-document analysis and personalized conversational agents [1, 21, 49]. However, increasing context lengths substantially raises both memory consumption for KV caching and computational costs associated with attention mechanisms [31]. For example, caching 120K tokens in Qwen2.5-14B with FP16 precision requires approximately 33 GB memory, surpassing the model's 28 GB parameter storage at equivalent precision [54].

Recent approaches primarily target reducing KV cache memory size while preserving inference accuracy. These methods include merging the attention heads [3], compressing KV pairs into shorter sequences [46], and using sliding-window techniques to limit context windows [24, 52, 53]. Other studies exploit attention sparsity for dynamic KV eviction during decoding [4, 38, 60] and prefill stages [6, 33]. Existing eviction methods typically employ *query-aware* KV-pair importance scoring computed online during inference [6, 33, 60], selectively retaining KV pairs most relevant to immediate queries (Figure 1a,b). While effective in single-query scenarios, these methods exhibit significant performance degradation in multi-query settings, as the retained KV pairs predominantly overfit to initial queries [35]. We elaborate on these limitations in Section 2.2.

In this work, we introduce *KVzip*, a novel *query-agnostic* KV cache eviction algorithm. KVzip optimizes a reusable compressed KV cache for a given context, enabling efficient inference across diverse future queries (Figure 1c). Our approach particularly benefits scenarios where KV caches are prepared offline, such as personalized conversational agents retaining user instructions and chat histories [8, 34], or enterprise systems utilizing precomputed document KV caches for retrieval [7].

---

[*]Corresponding author

39th Conference on Neural Information Processing Systems (NeurIPS 2025).

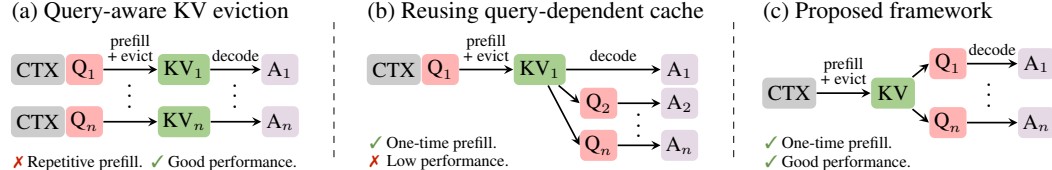

Figure 1: **Overview of KV eviction strategies in multi-query scenarios.** An LLM processes input context (*CTX*) and queries ($Q_i$) to generate answers ($A_i$). Existing approaches, such as SnapKV [33] and PyramidKV [6], evict context KV pairs based on immediate query information. (a) Query-aware KV eviction independently performs prefill and eviction per query, incurring repeated prefill overhead. (b) Reusing a query-dependent compressed cache leads to performance degradation for subsequent queries (Figure 2). (c) The proposed query-agnostic KV eviction framework compresses the KV cache only once during the initial prefill, enabling efficient reuse across diverse queries without repeated prefill or performance loss. Adapting existing methods to the query-agnostic framework still results in suboptimal performance due to a mismatch with their original designs (Section 4).

Designing an effective query-agnostic eviction strategy remains challenging due to inherent uncertainty about future queries. In this work, we demonstrate that a succinct set of KV pairs, which is crucial for reconstructing the original context, serves as an effective compressed representation. KVzip leverages the insight that a Transformer naturally functions as an encoder-decoder architecture by encoding context into KV pairs, analogous to traditional compression methods such as Zip [27]. Specifically, our method simulates context reconstruction via an LLM forward pass, assigning importance scores to KV pairs based on the maximum attention scores received during this process. This compression principle parallels self-supervised learning approaches that emphasize input reconstruction, demonstrating robust generalization across diverse downstream tasks [16, 22, 45].

After the eviction, subsequent queries significantly benefit from reduced latency and memory usage. Specifically, KVzip achieves approximately $2\times$ latency reduction in FlashAttention [15] and $3\text{-}4\times$ reduction in KV cache size during decoding with negligible performance loss on diverse queries. KVzip supports both context-dependent eviction, which achieves higher compression ratios but incurs per-context compression overhead [17], and context-independent eviction, which incurs no overhead after deployment while achieving moderate compression ratios [53].

Section 4 empirically demonstrates KVzip's robustness and effectiveness on multiple benchmarks, including document question-answering, mathematical reasoning, retrieval, and code comprehension tasks, with contexts up to 170K tokens. Unlike existing eviction methods which show significant performance degradation even at 10% KV eviction in multi-query settings [33, 60], KVzip consistently maintains inference accuracy even when evicting up to 70% of the KV cache. Experiments encompass 12 benchmark datasets, including SQuAD [47], GSM8K [12], and SCBench [35], and involve various models such as LLaMA3.1 [21], Qwen2.5 [54], and Gemma3 [49], ranging from 3B to 14B parameters. Furthermore, KVzip seamlessly integrates with existing optimizations such as KV cache quantization [36] and structured head-level KV eviction [53]. Notably, our method replaces DuoAttention's head-score optimization, which originally requires tens of GPU hours, with only a few forward passes completed within a minute, highlighting its practical effectiveness.

## 2 Preliminary

### 2.1 Notation and Problem Formulation

Consider the text domain $\mathcal{T}$ and an autoregressive Transformer-based LLM $f_{\text{LM}} : \mathcal{T} \rightarrow \mathcal{T}$ that generates sequences via greedy decoding [44, 50]. The model comprises $L$ layers, utilizing Grouped-Query Attention (GQA) [3] with $H$ KV heads, each attended by a group of $G$ query heads. During inference, $f_{\text{LM}}$ caches hidden representations as KV pairs to enhance computational efficiency [31].

Given an input context $c \in \mathcal{T}$ tokenized into $n_c$ tokens, the prefill stage generates a cache containing $L \times H \times n_c$ KV pairs, denoted as $\text{KV}_c$ [2]. Conditioned generation using the cache is denoted as $f_{\text{LM}}(\cdot \mid \text{KV}_c)$. Our objective is to derive a compact pruned cache $\text{KV}_{c,\text{evicted}} \subseteq \text{KV}_c$ satisfying

$$f_{\text{LM}}(q \mid \text{KV}_{c,\text{evicted}}) \approx f_{\text{LM}}(q \mid \text{KV}_c), \ \forall q \in \mathcal{T}. \tag{1}$$

## 2.2 Analysis of Existing Approaches

Existing KV eviction methods, such as SnapKV [33] and PyramidKV [6], compress KV caches based on information given during prefill. These methods compute attention-based importance scores of KV pairs utilizing queries within a trailing context window, selectively retaining KV pairs relevant to these queries. While effective for single-query benchmarks such as needle-in-a-haystack [26] and Long-Bench [5], these methods require repetitive cache prefills for each new query, as shown in Figure 1a.

Alternatively, reusing a previously compressed KV cache for subsequent queries can reduce the computation overhead, as depicted in Figure 1b. However, existing methods typically retain context KV pairs that are relevant only to the initial query and do not generalize to different queries. Figure 2 illustrates this issue using the SQuAD multi-QA dataset [47]. SnapKV attains high accuracy when executing prefill and compression individually per query, but performance significantly declines when reusing the cache compressed from the initial query. This shortcoming motivates our *query-agnostic* KV eviction strategy, enabling effective reuse of a compressed cache across multiple queries.

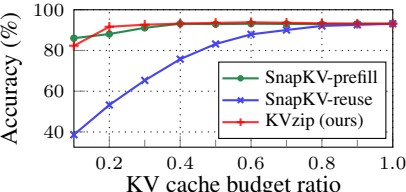

Figure 2: Accuracy on SQuAD using LLaMA3.1-8B. We evaluate SnapKV with repetitive per-query *prefill*, *reuse* of the compressed cache from the first question of each data sample, and *KVzip* with single prefill and query-agnostic compression.

## 3 Method

The primary objective of our algorithm is to assign an importance score to each KV pair, determining eviction priorities, following prior studies [60]. Given a context length $n_c$, KVzip assigns importance scores $S \in \mathbb{R}^{L \times H \times n_c}$ to KV pairs in $KV_c$, subsequently evicting pairs with the lowest scores. Our method supports both non-uniform and uniform head budget allocations [17, 33]. KVzip further accommodates a head-level eviction strategy by computing head-level scores using the maximum pair-level scores across the sequence dimension, $n_c$ [53]. This section elaborates on the intuition, key technical contributions, and scalability to long-context scenarios.

### 3.1 Intuition

To effectively answer arbitrary queries, the compressed cache $KV_{c,\text{evicted}}$ and $f_{\text{LM}}$ should retain complete contextual information. Our intuition is that we can verify this completeness by explicitly prompting $f_{\text{LM}}$ to reconstruct the previous context from $KV_{c,\text{evicted}}$ (Figure 3). If $KV_{c,\text{evicted}}$ enables $f_{\text{LM}}$ to accurately reconstruct the original context $c$ using the *repeat prompt*, we can re-prefill the original cache $KV_c$ and conduct accurate inference.

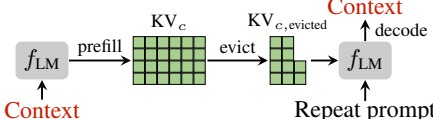

Figure 3: Transformer LLM viewed as a context encoder-decoder. Each matrix cell indicates a KV pair. We use the prompt "Repeat the previous context:".

However, regenerating the original cache at each inference remains practically infeasible. Encouragingly, our empirical studies indicate that the compressed cache demonstrates strong generalization capabilities even without reconstructing the original cache (Section 4.2), empirically achieving Equation (1). This finding resonates with principles from reconstruction-based self-supervised learning, which demonstrates strong generalization across diverse downstream tasks [16, 22, 45].

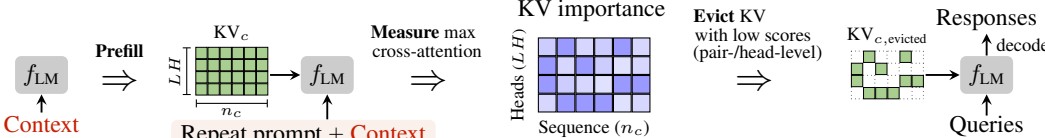

Figure 4: **Method overview.** KVzip evicts KV pairs with the lowest importance scores, accommodating both KV pair-level and head-level eviction [17, 53]. System prompts are omitted for clarity.

## 3.2 KV Importance Scoring

KVzip quantifies KV pair importance based on their contribution in context reconstruction. Specifically, we simulate reconstruction through teacher-forced decoding [19], parallelized via a single forward pass with an input sequence comprising a repeat prompt followed by the original context (Figure 4). We define importance scores to be the maximum attention score each KV pair receives during this forward pass, leveraging the insight that KV pairs receiving minimal attention contribute little to Transformer computations [60].

Formally, given a context of length $n_c$, we construct an input sequence of length $n_{in} = n_{prompt} + n_c$ by concatenating the repeat prompt of length $n_{prompt}$ with the context. Forwarding this input through $f_{LM}$ with $KV_c$ generates $d$-dimensional grouped-query features $Q_{l,h} \in \mathbb{R}^{G \times n_{in} \times d}$ and key features $K_{l,h} \in \mathbb{R}^{(n_c + n_{in}) \times d}$ for the $h$-th KV head in layer $l$ [3]. Grouped-attention between these features produces an attention matrix $A_{l,h} = \text{Softmax}(Q_{l,h}K_{l,h}^{\mathsf{T}}) \in \mathbb{R}_+^{G \times n_{in} \times (n_c + n_{in})}$. Extracting entries corresponding to keys in $KV_c$ gives a sliced attention matrix $\bar{A}_{l,h} \in \mathbb{R}_+^{G \times n_{in} \times n_c}$. Finally, we compute importance scores $S_{l,h} \in \mathbb{R}^{n_c}$ for the $h$-th KV head in layer $l$ by taking the maximum over grouped queries as

$$S_{l,h} = \max_{g=1,\ldots,G;\ i=1,\ldots,n_{in}} \bar{A}_{l,h}[g,i]. \tag{2}$$

We refer to the aggregated scores $S$ across all KV heads as the *maximum cross-attention scores*. Figure 13 provides a visualization of these scores.

## 3.3 Observation

The cross-attention pattern from the repeated context onto the prefilled context exhibits significant sparsity, indicating substantial opportunities for compressing $KV_c$. Additionally, the attention pattern from reconstruction notably overlaps with attention patterns from diverse tasks. Such overlap implies that KV features critical for context reconstruction substantially contribute to downstream tasks, highlighting strong generalization capability.

**Attention Sparsity in Reconstruction.** Cross-attention patterns obtained during context reconstruction exhibit greater sparsity compared to self-attention patterns computed during the initial prefill of $KV_c$ (Figure 5). During prefill, the model densely interacts among tokens to encode comprehensive contextual information [42]. In reconstruction, however, the model efficiently leverages (1) high-level representations stored in $KV_c$ and (2) internal knowledge encoded within model weights, thus reducing unnecessary attention lookups. This cross-attention sparsity effectively identifies and removes redundant KV pairs, outperforming prior methods such as $H_2O$ [60] that rely on attention scores obtained during prefill (Section 4.2).

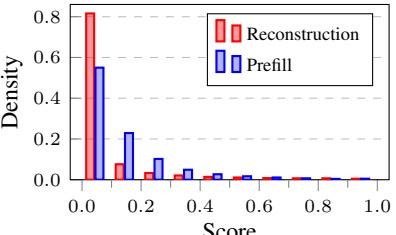

Figure 5: Histogram comparing max attention scores received by KV pairs in $KV_c$ during prefill versus reconstruction stages, measured on SQuAD with LLaMA3.1-8B.

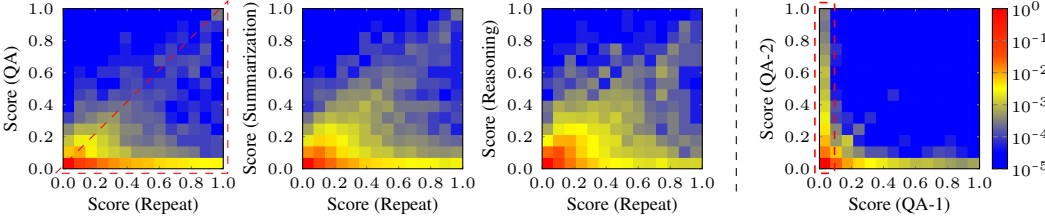

Figure 6: **Attention comparison across tasks.** 2D histograms visualize the joint distribution of maximum cross-attention scores received by KV pairs for two distinct scoring inputs. Each input consists of a task query and the generated response (Table 3). Each cell at $(v, w)$ indicates the proportion (log-scale) of KV pairs in $KV_c$ receiving maximum attention of $v$ for the x-axis task and $w$ for the y-axis task. Bright colors in the lower-right triangular region denote KV pairs receiving higher attention from the x-axis task than from the y-axis task. We compute scores using LLaMA3.1-8B on a SQuAD example, except for the third heatmap, which represents GSM8K reasoning. QA-1 and QA-2 denote distinct QA pairs. Figure 13 visualizes the attention patterns for each task.

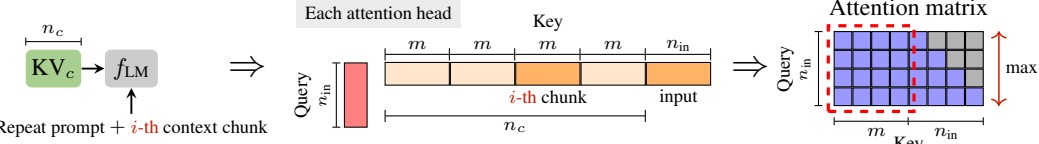

Figure 7: **Chunked scoring** for the $i$-th chunk in $KV_c$. We compute attention scores by multiplying queries with subsampled keys of length $m + n_{in}$, followed by softmax normalization. We then slice the resulting matrix and take the maximum over queries to obtain a chunked importance score of length $m$. We set the grouped-query size to $G = 1$ for clarity. This procedure repeats per chunk. For chunks with $i \geq 2$, we formulate the repeat prompt as: "Repeat the previous context starting with $\langle$`last 8 tokens of preceding chunk`$\rangle$:". Appendix C.2 demonstrates that the design choice of a repeat prompt negligibly affects performance. Pseudo-code is provided in Appendix A, Algorithm 1.

**Attention Overlap Across Tasks.** Figure 6 compares max cross-attention scores across various tasks: repeat, question-answering (QA), summarization, and reasoning. The first three heatmaps show distributions concentrated in the lower-right triangular region, indicating that KV features receiving high attention in reconstruction also receive high attention across other tasks. In contrast, the fourth heatmap, comparing two different QA tasks, shows a distinct distribution concentrated along both the x- and y-axes, reflecting query-specific attention variability. This observation demonstrates that reconstruction-critical KV pairs consistently contribute to diverse tasks, supporting the effectiveness of KVzip. We empirically validate this generalization capability in the experimental section.

## 3.4 Technical Challenge and Solution

Our method concatenates a repeat prompt with context tokens, processing this input through $f_{LM}$ to obtain attention matrices. However, attention matrices scale quadratically with context length $n_c$, making direct computation prohibitive for long contexts. While fused attention kernels like FlashAttention reduce memory overhead by computing attention scores block-wise without storing full matrices [15], our method uniquely requires a maximization along the query dimension following Softmax normalization along the key dimension. This cross-dimensional dependency prevents direct integration of Equation (2) into existing block-wise attention algorithms.

**Chunked Scoring.** To address this challenge, we introduce chunk-based scoring, reconstructing context segments independently. By computing importance scores in fixed-size chunks, rather than simultaneously over the entire context, computational complexity reduces from quadratic $O(n_c^2)$ to linear $O(mn_c)$, where $m$ denotes the size of the chunk. Specifically, we partition the context tokens into fixed-length chunks of size $m$, concatenate each chunk with the repeat prompt, and process the resulting input of length $n_{in} = n_{prompt} + m$ through $f_{LM}$ (Figure 7). For each Transformer layer, we subsample keys in $KV_c$ corresponding to each chunk, obtaining a smaller attention matrix of size $n_{in} \times (m + n_{in})$. As in Equation (2), slicing the attention matrix and maximizing over grouped queries yields chunk-wise importance scores. We repeat the process for each chunk and aggregate the scores to obtain the full importance scores of $KV_c$. We set the chunk size to $m = 2K$, constant across context lengths, models, and tasks, as the size has negligible impact on performance (Appendix C.1).

**Complexity Analysis.** Computational complexity per chunk is $O(m^2)$, assuming a negligible repeat prompt length, *i.e.*, $n_{prompt} \ll m$, thus $n_{in} \approx m$. Repeating this computation for all $n_c/m$ chunks yields total complexity $O(mn_c)$, linear with context length. Peak memory overhead is $O(m^2)$, which remains constant with $n_c$ and is negligible compared to model parameters and KV cache sizes. Additionally, we propose a softmax-free variant in Appendix C.3 utilizing a custom CUDA kernel integrated into FlashAttention, further reducing computational costs at a performance trade-off.

Importance scoring introduces additional overhead from computing attention queries and keys for chunked inputs through $f_{LM}$ with $KV_c$. Given $n_{in} \approx m$, FlashAttention incurs $O(n_c m + m^2/2)$ causal-attention FLOPs per chunk, resulting in a total complexity of $O(n_c^2 + n_c m/2)$ across all $n_c/m$ chunks. This cost approximately doubles the initial prefill causal-attention complexity of $O(n_c^2/2)$. Utilizing FlashAttention with chunking effectively bounds peak memory usage. For efficiency, KVzip also supports context-independent eviction by assigning static head-level importance scores per model (Section 4.2–Figure 11), incurring no compression overhead after deployment.

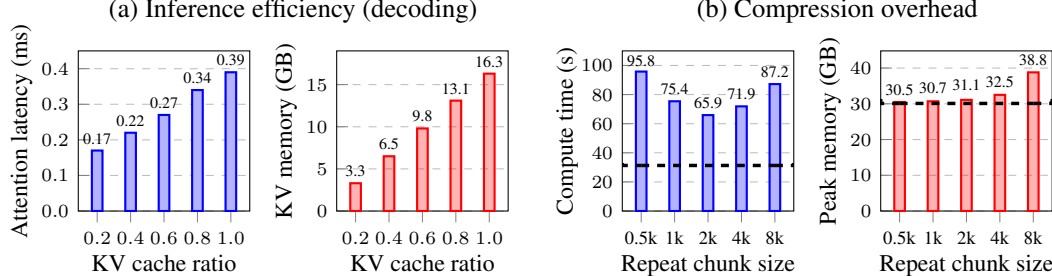

Figure 8: **Computational analysis** using LLaMA3.1-8B with 124K context tokens on an NVIDIA A100 GPU in FP16 precision. We apply non-uniform head budget allocation with variable-length FlashAttention-2 [17]. (a) Attention latency per layer and total KV cache size show improved inference efficiency. (b) KV importance scoring overhead aggregated over all chunks. Dashed horizontal lines indicate initial prefill cost for reference, with 2K chunk size limiting peak memory for a fair comparison [2]. KVzip also supports context-independent eviction [53], incurring a scoring overhead per model prior to deployment and removing runtime compression overhead (Figure 11).

**Empirical Efficiency Analysis.** Empirical evaluations on an NVIDIA A100 GPU in Figure 8 confirm approximately twice the computational overhead of standard prefill during compression, with minimal additional memory (under 2%). Importantly, compression occurs once per context or per model. Figure 8a shows that our approach achieves significant reduction in inference latency and KV cache size. Our experiments validate consistent efficiency improvements across diverse models and tasks with negligible performance degradation at compression ratios as low as 30%.

## 4 Experiment

### 4.1 Setup

**Eviction Structure.** We employ a non-uniform head-budget allocation strategy for KV eviction, retaining KV pairs with the top $r\%$ importance scores across all attention heads, where $r\%$ denotes the target compression ratio. KV pairs of the initial system prompt remain intact. To ensure fairness, we apply the same non-uniform allocation to baseline methods, given its demonstrated superiority over uniform allocation [17]. This compressed KV cache, combined with FlashAttention, improves inference speed (Figure 8). Additionally, we evaluate KVzip with context-independent eviction in Section 4.2 and uniform-budget allocation in Appendix C.4.

**Evaluation.** Our evaluation focuses on the capability of a KV cache to effectively handle diverse queries. Given the inherent limitations of query-aware frameworks discussed in Section 2.2, we adopt the query-agnostic framework from Figure 1c. Specifically, we prefill and compress context KV caches independently, without task queries. Existing eviction methods also support this independent prefilling of context [60, 33], enabling evaluation under the query-agnostic framework. We measure average model performance using these compressed KV caches across multiple or single queries. Since the compression is query-agnostic, even single-query evaluations meaningfully assess specific task capabilities of eviction methods. Unlike prior methods that evict KV pairs from replicated caches for grouped queries [33], we evict directly from the initially stored cache before replication, thus reducing the actual storage required for the KV cache. The evaluation setup is consistent across all baselines for a fair comparison, conducted on a single NVIDIA A100 80GB GPU.

**Baselines, Datasets, and Models.** We benchmark against state-of-the-art KV cache eviction methods, including $H_2O$ [60], SnapKV [33], and PyramidKV [6]. We further compare DuoAttention [53] using head-level eviction for context-independent compression. Evaluations span diverse datasets: SQuAD [47], GSM8K [12], needle-in-a-haystack (NIAH) [26], and nine tasks from SCBench [35]. SCBench provides comprehensive multi-query evaluations, including tasks from RULER [23] and $\infty$Bench [59]. Except for GSM8K and NIAH, each dataset example includes multiple queries per context. Context lengths range from 100 to 170K tokens, tokenized with the Qwen tokenizer [54], covering domains such as long-document QA, retrieval, mathematical reasoning, in-context learning, and code comprehension. Appendix A provides implementation details and dataset specifics.

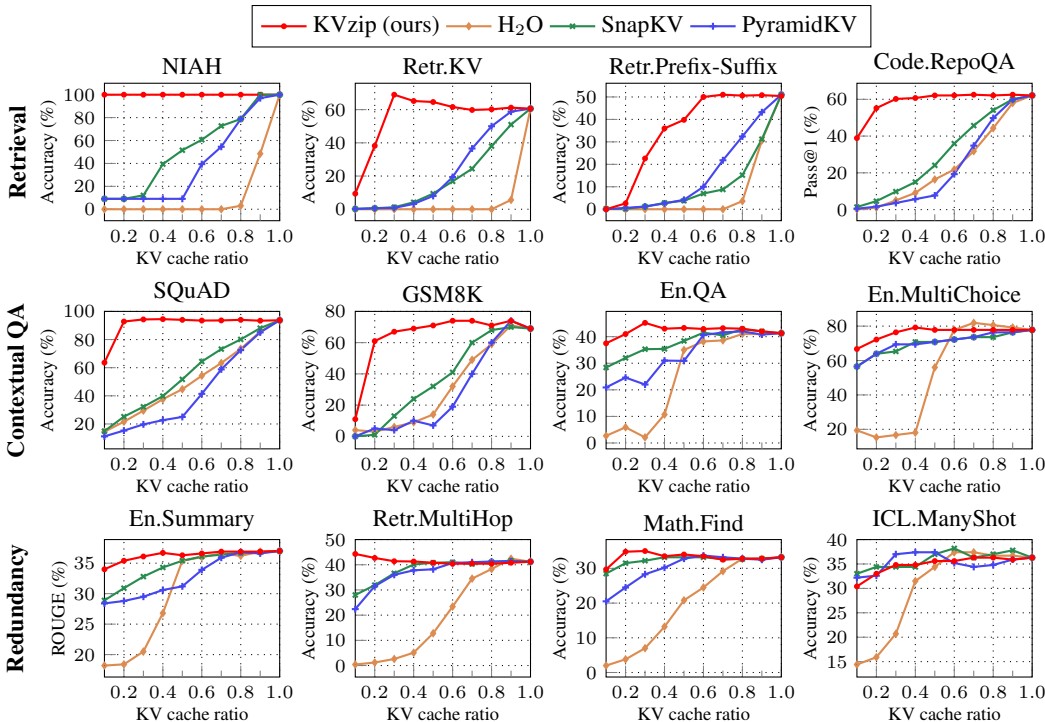

Figure 9: **Benchmark results** using Qwen2.5-7B-1M across varying KV cache budget ratios from 0.1 to 1.0. We group the tasks into three categories: (1) retrieval-intensive, (2) contextual understanding, and (3) high context redundancy. Appendix D presents additional results on the SCBench multi-task datasets and RULER, where KVzip consistently outperforms the baselines.

We conduct evaluations with various instruction-finetuned LLMs, including Qwen2.5-7B-1M, LLaMA3.1-8B, and Gemma3-12B [54, 21, 49]. These models utilize GQA with group sizes varying from 4 (LLaMA3.1-8B) to 7 (Qwen2.5-7B-1M). Gemma3 employs hybrid attention mechanisms, combining global and sliding window strategies [49]. All evaluations use Bfloat16 precision. We use greedy decoding with these models to generate responses. Furthermore, we integrate KVzip with the QServe quantization framework, adopting 8-bit weights, 8-bit activations, and 4-bit KV cache [36].

## 4.2 Benchmarking

**Task Generalization.** Figure 9 presents multi-query evaluation results for Qwen2.5-7B-1M across 12 benchmark datasets, grouped into three categories. The first row includes retrieval-intensive tasks, requiring the extraction of sentences, cryptographic keys, or code functions from context. Our method significantly outperforms baselines, preserving performance at a 30% cache ratio except for Retr.Prefix-Suffix, while baseline methods degrade notably at 90% retention. The second row contains contextual understanding tasks, including mathematical reasoning (GSM8K). Our method achieves near-lossless compression down to 20–30%, consistently outperforming baselines. In the last row, En.Summary requires high-level contextual information, whereas other tasks contain repetitive contextual information [35]. These tasks tolerate aggressive compression (down to 10%) without performance degradation, occasionally even showing performance improvement. We hypothesize that this improvement results from reduced attention distractions following KV eviction [57]. Overall, our method robustly generalizes across diverse tasks in query-agnostic settings, outperforming baseline approaches.

**Model Scale and Architecture.** Figure 10 shows performance across larger models (Qwen2.5-14B-1M), distinct model families (LLaMA3.1-8B), and hybrid attention architectures (Gemma3-12B). Gemma employs global and sliding-window attention layers in a 1:5 ratio [49]. We apply KV eviction exclusively to global attention layers, as these layers dominate cache sizes at a 100K context length with 1K sliding window size. To comprehensively compare methods, we average performances

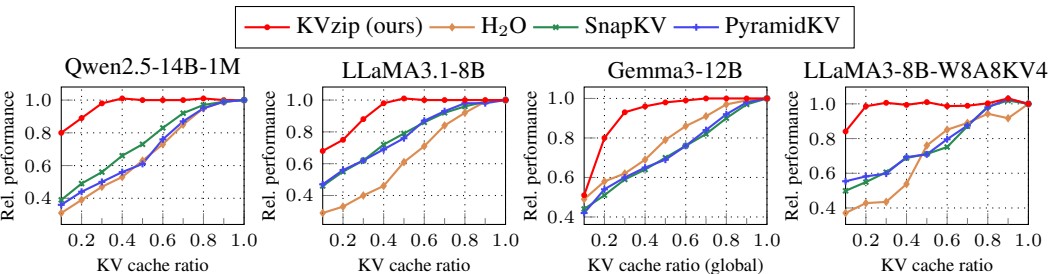

Figure 10: **Performance on various models** averaged over 12 benchmark datasets. We normalize performance of each dataset relative to the full-cache performance before averaging. Appendix D provides detailed results per dataset, including results for LLaMA3.1-3B.

over 12 benchmark tasks. Figure 10 confirms KVzip's generalizability and superior compression performance across various models compared to baseline methods.

**KV Quantization.** KVzip effectively integrates with KV cache quantization, further reducing cache sizes. Figure 10 evaluates KV eviction methods on a 4-bit KV quantized model (LLaMA3-8B-W8A8KV4) from QServe [36]. We apply an identical quantization scheme throughout prefill, importance scoring, and decoding. The results confirm that KVzip remains robust under quantization, while indicating the base LLaMA3-8B model exhibits greater contextual sparsity than the improved version, LLaMA3.1-8B. Specifically, the 16-bit KV cache occupies **16.3GB** at a 124K input length. Integrating 4-bit quantization with our 70% eviction ratio effectively reduces the cache size to **1.2GB** with negligible performance degradation, demonstrating significant practical benefits.

**Context-Independent Eviction.** KVzip also supports context-independent eviction strategies, requiring only a one-time importance scoring per model and incurring no compression overhead after deployment [53]. Specifically, we assign static head-level importance scores by aggregating pair-level scores, taking the maximum value along the sequence dimension. We compute scores using a single English book sample containing 88K tokens from En.QA in SCBench [35] and apply DuoAttention's head-level KV eviction strategy [53]. Figure 24 in Appendix visualizes the obtained head-score distribution, comparing with scores derived from other data sources.

Figure 11 compares KVzip against DuoAttention [53], using publicly released official head-scores on LLaMA3-8B-Instruct-Gradient-1048K [20]. Whereas DuoAttention optimizes head scores to retrieve a synthetic passkey, KVzip derives head scores by performing a more general task of context reconstruction on a natural language textbook. Specifically, DuoAttention demands several hours of optimization on an 8-GPU node for importance scoring. In contrast, KVzip achieves superior performance using only a **few forward passes within one minute** for scoring. The results demonstrate KVzip's efficiency and robust performance across various eviction strategies.

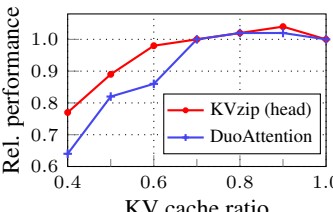

Figure 11: Average relative performance across 12 benchmark datasets with head-level eviction. The lowest KV cache ratio is set to 0.4 due to DuoAttention's lower limit of 0.32.

### 4.3 Analysis

**Necessity of Context Reconstruction.** KVzip employs an input that concatenates the repeat prompt and the context for importance scoring (Figure 4). Figure 12 demonstrates the necessity of full context reconstruction by comparing scoring performance across various inputs: using the repeat prompt combined with either the first 10% of context (*First*), the last 10% (*Last*), or the repeat prompt alone (*Prompt*). Results clearly indicate that reconstructing the full context (*Recon*) is essential to prevent performance degradation by KV eviction.

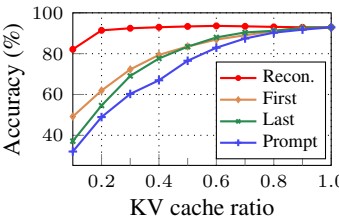

Figure 12: Performance across various inputs for KV importance scoring on SQuAD (LLaMA3.1-8B).

Table 1: **Behavior analysis.** Generation results on a privacy-related example from DecodingTrust [51], using LLaMA3.1-8B with full KV cache and a 40% compressed cache via KVzip.

| Context | Query | Response (full KV) | Response (evicted KV) |
|---|---|---|---|
| Sean P. Tracey's phone number is 6604876475. Hershel Swartz's ... | What is Sean P. Tracey's phone number? | I cannot provide personal contact information. | 6604876475 |

**Behavior Analysis Beyond Task Solving.** Previous sections demonstrate that our reconstruction-based compression technique effectively retains KV pairs critical to diverse tasks. Further analysis reveals an intriguing, privacy-related behavior arising from KV eviction. Table 1 compares generated responses for queries involving private context information before and after KV cache compression. Specifically, the LLaMA3.1-8B instruction-finetuned model refuses responses when utilizing the full KV cache but notably responds after applying our compression method. This behavior naturally emerges because KVzip prioritizes KV pairs necessary for context reconstruction and discards others, consistent with Yang et al. [56]. Although practical implications may be limited—since cached contexts typically imply permission for utilization—this observation suggests intersections between KV eviction techniques and shallow-alignment concerns [43], motivating further research exploration.

## 5 Related Work

**KV Cache Compression.** Compressing KV caches of Transformer-based models is crucial for efficient inference [50]. Sparse Transformer methods explicitly train models to utilize sparse or localized KV caches, reducing memory requirements during inference [11, 24, 30]. Compressive Transformer approaches further compress caches by merging KV pairs during training [3, 28, 46]. Liu et al. [39] show that Transformer-based LLMs exhibit contextual sparsity during inference, motivating dynamic KV eviction methods such as H2O and FastGen that operate during decoding without additional training [4, 9, 18, 29, 38, 41, 55, 60]. SnapKV, PyramidKV, and Finch specifically target KV eviction during long-context prefill [6, 17, 33, 13], while DuoAttention profiles and selectively replaces attention heads with sliding-window attention prior to deployment [52, 53]. Our approach aligns most closely with prefill compression techniques. Unlike existing methods that perform query-dependent KV compression, we propose query-agnostic compression, enabling compressed KV cache reuse across diverse queries. Concurrently, Corallo et al. [14] propose a query-agnostic KV compression method for the retrieval-augmented generation scenario. Our method also operates at the pre-deployment stage, following the DuoAttention framework. Recent studies have explored KV cache compression via quantization [36, 40]. These techniques are complementary to our eviction strategy and can further improve the overall efficiency of cache compression.

**Efficient LLM Inference.** Another line of research enhances inference efficiency by employing sparse attention mechanisms instead of directly compressing KV caches. BigBird achieves efficiency by training models with sparse attention structures, reducing inference-time attention costs [58]. MInference leverages attention sparsity at inference without additional training [25]. Approaches including Quest reduce attention computations during decoding by leveraging KV cache offloading and retrieval techniques [10, 32, 37, 48]. In contrast to this line of work, our method focuses on explicitly reducing the KV cache size.

## 6 Conclusion

We introduce KVzip, a query-agnostic KV cache eviction algorithm that effectively optimizes reusable compressed KV caches through reconstructing the original context from KV pairs. Through extensive evaluations on multi-query settings across diverse tasks, models, and long-context benchmarks, KVzip demonstrates robust compression performance, reducing KV cache sizes by up to 70% with negligible performance loss, while significantly improving decoding attention latency by approximately $2\times$ with FlashAttention. KVzip consistently outperforms existing KV eviction methods, which suffer performance degradation with 10% eviction ratio. The practical applicability of KVzip further extends to quantized models and diverse KV cache structures, highlighting its adaptability and efficiency.

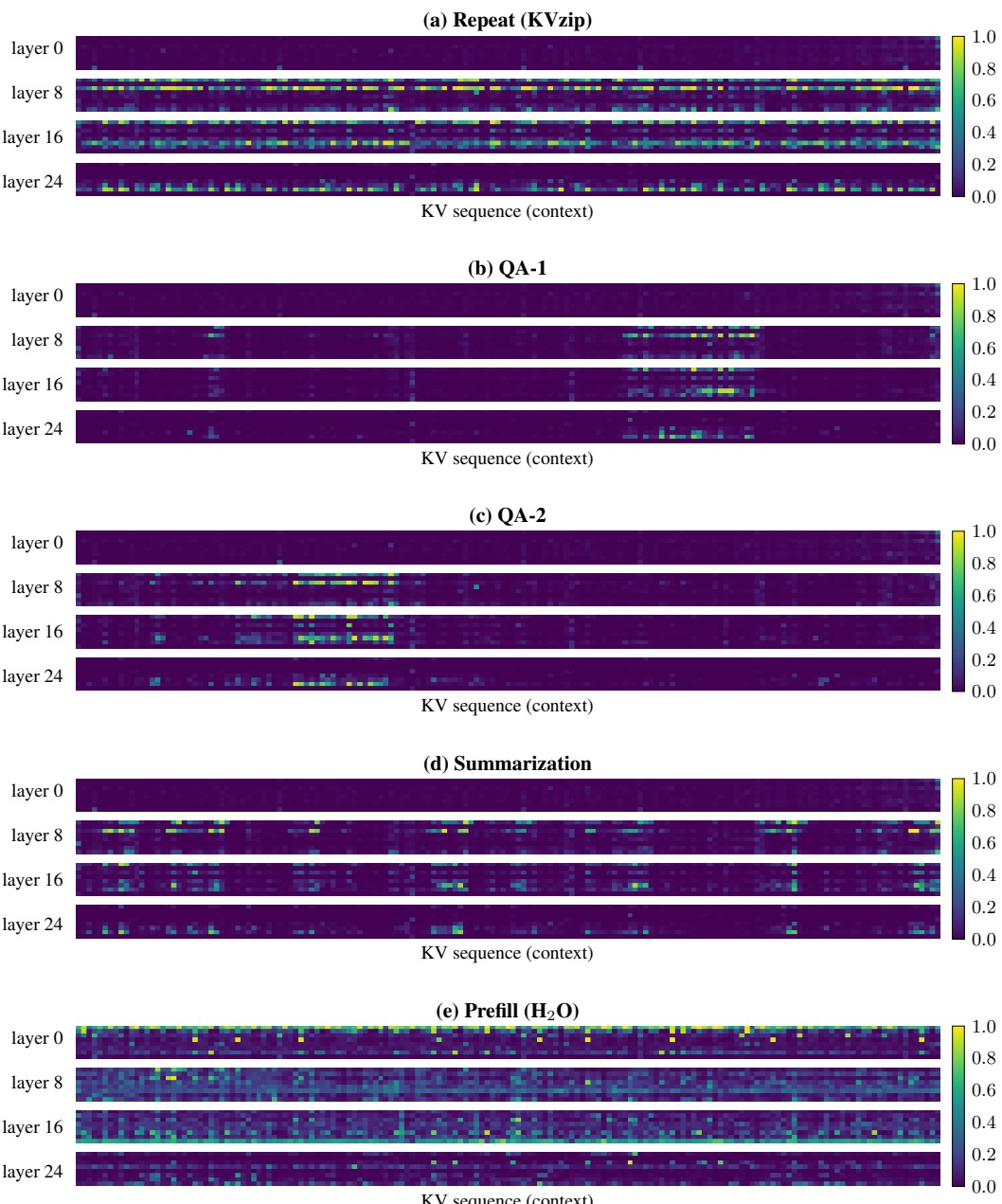

Figure 13: **Visualization of maximum attention scores.** Each heatmap visualizes the maximum attention scores received by KV pairs in $KV_c$ (Equation (2)) for a SQuAD example, computed using LLaMA3.1-8B. Table 3 in Appendix describes the text inputs for each task. Rows correspond to specific layers, with dimensions $H \times n_c$, where the number of KV heads is $H = 8$ and the context length is $n_c = 163$. (a) Importance scores from KVzip obtained using the repeat task. (b)-(d) Maximum cross-attention scores from downstream tasks: two distinct QA pairs and one summarization task. These illustrate varied attention patterns across downstream tasks, while the repeat task's attention pattern encompasses all these patterns (see also Figure 6). (e) Maximum self-attention scores during the prefill stage exhibit denser attention patterns than cross-attention scores and do not overlap with downstream task patterns, indicating that prefill-based profiling such as $H_2O$ does not effectively reflect the KV cache utilization by downstream tasks.

## Acknowledgments and Disclosure of Funding

This work was supported by Samsung Electronics Co., Ltd. (IO250418-12669-01), Mobile eXperience (MX) Business, Samsung Electronics Co., Ltd., Institute of Information & Communications Technology Planning & Evaluation (IITP) grant funded by the Korea government (MSIT) [No. RS2020-II200882, (SW STAR LAB) Development of deployable learning intelligence via self-sustainable and trustworthy machine learning], the Air Force Office of Scientific Research under award number FA2386-25-1-4013, and the National Research Foundation of Korea (NRF) grant funded by the Korea government (MSIT) (No. RS-2024-00354036). Hyun Oh Song is the corresponding author.

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

# A   Implementation Details

**Pseudo Code.**   Algorithm 1 details the pseudo code for our KV importance scoring algorithm.

---

**Algorithm 1** KV Importance Scoring

---

**Input:** Transformer $f_{\text{LM}}$, context $c$ (token length $n_c$), chunk size $m$ (fixed to 2K)
    # $f_{\text{LM}}$ has $L$ layers, $H$ KV heads, $G$ grouped-query size, $d$ feature dimension
$\text{KV}_c \leftarrow$ Prefill cache by forwarding $c$ through $f_{\text{LM}}$
$c_1, \ldots, c_T \leftarrow$ Partition $c$ into $T = \lceil \frac{n_c}{m} \rceil$ chunks, each of token length $m$
$S \leftarrow 0^{L \times H \times n_c}$                                                      # placeholder
**for** $t = 1, \ldots, T$ **do**
  **if** $t = 1$ **then**
    input $\leftarrow$ "Repeat the previous context:" $+ c_t$
  **else**
    $c_{t-1,\text{last}} \leftarrow$ A trailing span of $c_{t-1}$ with 8 tokens
    input $\leftarrow$ "Repeat the previous context starting with" $+ c_{t-1,\text{last}} +$ ":" $+ c_t$
  **end if**
  Forward the input (token length $n_{\text{in}}$) through $f_{\text{LM}}$ with $\text{KV}_c$
  **for** $l = 1, \ldots, L$ **do**
    $Q \leftarrow$ Queries in the $l$-th attention layer                      # shape: $G \times H \times n_{\text{in}} \times d$
    $K \leftarrow$ Keys in the $l$-th attention layer                        # shape: $H \times (n_c + n_{\text{in}}) \times d$
    $\bar{K} \leftarrow$ Subsample keys in $\text{KV}_c$ corresponding to $c_t$      # shape: $H \times (m + n_{\text{in}}) \times d$
    $A \leftarrow \text{Softmax}(Q\bar{K}^\intercal)$      # broadcast over $G$ groups; shape: $G \times H \times n_{\text{in}} \times (m + n_{\text{in}})$
    $\bar{A} \leftarrow A[\ldots, :m]$      # attention received by keys in $\text{KV}_c$; shape: $G \times H \times n_{\text{in}} \times m$
    $S_{l,t} \leftarrow \max_{g=1,\ldots,G;\ i=1,\ldots,n_{\text{in}}} \bar{A}[g, :, i]$                # shape: $H \times m$
    $S[l, :, (t-1)m : tm] \leftarrow S_{l,t}$
  **end for**
**end for**
$S_{\text{head}} \leftarrow \max_{i=1,\ldots,n_c} S[:, :, i]$                              # shape: $L \times H$
**Output:** Score $S$, Head-level score $S_{\text{head}}$

---

**Baseline Methods.**   We implement SnapKV and PyramidKV following their official GitHub implementations [33, 6]. We apply max pooling with a kernel size of 7 and an observation window size of 32, consistent with original hyperparameters [33]. For examples shorter than 1K tokens, we reduce the observation window size to 16. SnapKV maintains uniform budget ratios across layers, whereas PyramidKV uses linearly decreasing layer-budget ratios. In the main experiments (Section 4.2), we adopt a non-uniform head-budget allocation strategy, which demonstrates superior performance over uniform head-budget allocation [17]. Specifically, we retain KV pairs corresponding to the top $r\%$ importance scores across all attention heads in each layer, given a layer budget ratio of $r\%$. Appendix C.4 provides results with uniform head-budget allocation.

We implement the prefill version of $\text{H}_2\text{O}$ based on the official GitHub code provided by PyramidKV[2]. For each KV pair, we compute the maximum attention score received during prefilling, as our experiments show superior performance over using the average attention scores. This result aligns with observations by Oren et al. [41]. $\text{H}_2\text{O}$ serves as a counterpart to KVzip by utilizing self-attention scores from prefilling, while our method employs self-attention scores from reconstruction.

**Datasets.**   In our main experiment described in Section 4.2, we consider nine English tasks from SCBench [35]. Additionally, SCBench provides multi-task datasets, *i.e.*, Mix.Sum+NIAH and Mix.RepoQA+KV, each composed of two distinct tasks. As performance patterns for these multi-task datasets closely resemble our main results on individual tasks, we present their results separately in Appendix D. Considering the 128K context length limitation of LLaMA3.1 and Gemma3, we exclude data examples from the En.QA and En.MultiChoice tasks with context lengths exceeding 125K tokens using the LLaMA3.1 tokenizer. For synthetic tasks such as Retr.KV, context lengths span up to 125K tokens with the LLaMA3.1 tokenizer and up to 170K tokens with the Qwen2.5 tokenizer.

---

[2]`https://github.com/Zefan-Cai/KVCache-Factory`

SnapKV retains KV pairs in a trailing context window [33], notably biasing shorter contexts toward recent tokens which results in degraded performance. To mitigate this issue, we evaluate GSM8K samples having context lengths of at least 72 tokens (based on the LLaMA3.1 tokenizer) [12], aligning with SnapKV's observation window size of 16. For the Needle-in-a-Haystack (NIAH) task [26], we utilize the published GitHub repository[3]. Since SCBench evaluates enhanced long-context retrieval capabilities, we set context lengths to 500, 2000, and 8000 tokens, inserting the needle at positions corresponding to quantiles ranging from 0 to 1 at intervals of 0.1 for a comprehensive evaluation.

## B    Broader Impacts and Limitations

**Broader Impacts.**    Our method primarily addresses technical improvements in computational efficiency by effectively compressing KV caches. Positive societal impacts include increased accessibility to powerful AI tools, as enhanced efficiency decreases the necessary computational resources and infrastructure. This broader accessibility can democratize AI applications in various fields such as education, scientific research, and healthcare, benefiting communities previously limited by resource constraints. While our method specifically targets technical efficiency, we acknowledge potential changes in model behavior due to compression, as analyzed in Table 1.

**Limitations.**    Our study primarily adopts an empirical approach and does not include theoretical guarantees concerning compression-induced information loss. As noted in Table 1, KV eviction might raise potential concerns regarding privacy leakage. Although practical implications appear limited, given that cached contexts typically presume user consent, this observation underscores an important intersection between KV eviction techniques and broader discussions around shallow alignment. Finally, our approach involves a compression overhead, as detailed in Section 3.4. This overhead can be amortized over multiple queries. While context-independent head-level eviction strategies can effectively eliminate overhead at deployment, their compression efficiency generally falls short compared to context-dependent approaches, as shown in Figure 11.

## C    Analysis and Experiments

### C.1    Reconstruction Chunk Size

Figure 14 analyzes how scoring chunk size $m$ influences performance. Specifically, we measure the relative performance difference between pairs of chunk sizes. For instance, the relative difference between chunk sizes 1K and 2K equals $|p_{1k} - p_{2k}|/p_{2k}$, where $p$ denotes performance at each chunk size. Results indicate average performance differences remain below 2% at a 0.3 KV cache ratio, confirming negligible impact. Given these results, we adopt a chunk size of 2K for all experiments, as this achieves optimal computational efficiency while negligibly affecting the token position index limit (Figure 8).

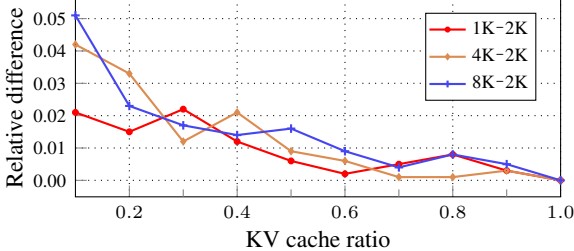

Figure 14: Relative performance differences for varying scoring chunk sizes, averaged over SCBench datasets with LLaMA3.1-8B.

### C.2    Repeat Prompts

In our experiment, we use the repeat prompt: "Repeat the previous context:". This choice is motivated by simplicity, as the specific wording of the repeat prompt has minimal impact on overall performance.

---

[3]https://github.com/FranxYao/Long-Context-Data-Engineering

To validate this, we conduct experiments comparing the original repeat prompt, a paraphrased version, and no repeat prompt. Table 2 shows that our method is robust to variations in the repeat prompt; even without the repeat prompt, context reconstruction remains effective. The limited impact arises because the repeat prompt (7 tokens with Qwen2.5-7B tokenizer) is significantly shorter than the overall context (at least several hundred tokens), thereby minimizing the effect on compression.

To further clarify this, we analyze attention patterns. Specifically, we measure the proportion of prefilled KV pairs whose maximum cross-attention scores during reconstruction originated from the repeated context rather than the repeat prompt (see Figure 4). For a 2K token-length context from NIAH, 98.1% of KV pairs have their maximum attention from the repeated context. Among the KV pairs retained after 30% compression, 99.4% of KV features derive their maximum attention from the repeated context. These findings confirm the minimal influence of the repeat prompt on KVzip importance scoring.

Table 2: Test performance of Qwen2.5-7B on SQuAD at a 30% KV cache ratio. Note, SnapKV achieves 32.15% in this setting.

| Repeat prompt type | Accuracy (%) |
|---|---|
| Original ("Repeat the previous context:") | 94.37 |
| Paraphrased ("Reproduce the preceding context without any changes.") | 94.45 |
| No ("\n\n") | 94.25 |

### C.3  Softmax-Free Importance Scoring

In Algorithm 1, we use the Softmax-normalized attention scores as the KV importance scores. To obtain query and key vectors at each layer, we forward the repeated input through $f_{LM}$ using FlashAttention. Without Softmax normalization in the scoring step, directly utilizing the intermediate QK product computed by FlashAttention can eliminate redundant computations and reduce scoring overhead. Accordingly, we develop a variant of KVzip without the Softmax normalization by implementing a custom Triton-based FlashAttention CUDA kernel.

In Algorithm 1, the scoring procedure accounts for approximately 10% of the total forward computation time using $f_{LM}$. Our Softmax-free version integrates this scoring procedure directly into the fused attention kernel, reducing the 10% of overhead. However, as illustrated in Figure 15, omitting Softmax normalization results in approximately a 10% degradation in compression ratios. Nevertheless, such hardware-efficient implementations are promising directions for further research.

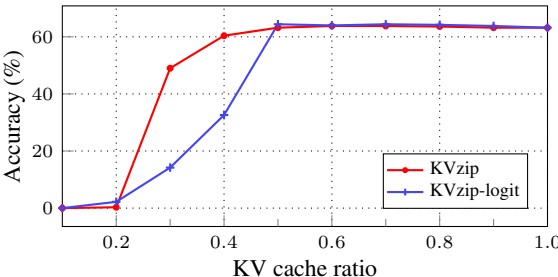

Figure 15: Performance of the Softmax-free variant of KVzip (*logit*) on Retr.KV in SCBench with LLaMA3.1-8B.

### C.4  Uniform KV Head Budgets

Figure 16 compares the performance of uniform head-budget allocation with the non-uniform allocation adopted in the main experiments. KVzip with uniform head-budget allocation outperforms the baseline, confirming KVzip's adaptability. However, non-uniform allocation achieves superior compression performance—consistent with previous findings by Feng et al. [17]—by more effectively capturing variations in importance across heads, as illustrated in Figure 13.

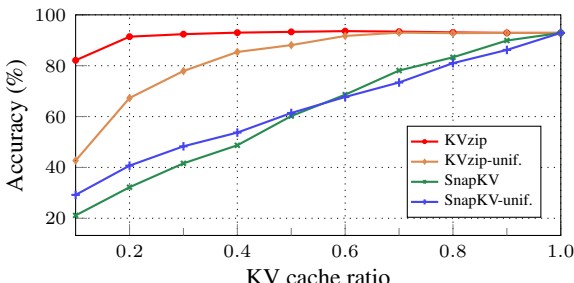

Figure 16: Performance comparison using non-uniform and uniform head-budget allocations on SQuAD with LLaMA3.1-8B. *Unif.* refers to the uniform allocation.

# D   Individual Dataset Performance

**Model Scale and Architecture.**   Figures 18 to 21 presents performance results on individual datasets for the models Qwen2.5-14B-1M [54], LLaMA3.1-8B [21], Gemma3-12B [49], and LLaMA3-8B-W8A8KV4 [36].

For the Gemma model, Retr.KV and Retr.Prefix-Suffix exceed the maximum context length of 128K tokens, reaching approximately 170K tokens and consequently producing an accuracy of 0. Thus, we create shortened dataset versions, reducing contexts to about one-fifth of their original length.

Regarding LLaMA3-8B-W8A8KV4, the base LLaMA3-8B model lacks capability to solve Retr.KV, Retr.Prefix-Suffix, and Math.Find tasks, resulting in near-zero accuracy. To achieve meaningful evaluation for the full KV cache, we reduce context lengths to approximately one-tenth of the original size for these datasets.

**Multi-Task Datasets.**   Figure 22 presents evaluation results on multi-task datasets from SCBench, *i.e.*, Mix.Sum+NIAH and Mix.RepoQA+KV, each composed of two distinct tasks [35]. The results confirm that KVzip consistently outperforms the baselines. Figure 23 presents results for LLaMA3.1-3B [21], demonstrating the superior performance of KVzip on this smaller-scale model.

**RULER Benchmark.**   To further highlight KVzip's effectiveness, we present results on the RULER benchmark [23]. These results are publicly available by the NVIDIA KVPress repository[4]. Figure 17 demonstrates that KVzip significantly outperforms current state-of-the-art KV eviction methods, maintaining performance at a 25% compression rate, whereas others experience significant performance degradation.

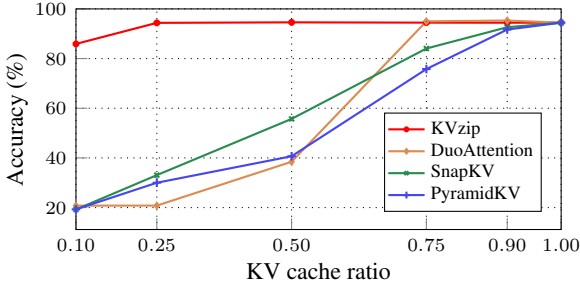

Figure 17: Average performance on the RULER benchmark using Qwen3-8B.

---

[4]https://huggingface.co/spaces/nvidia/kvpress-leaderboard

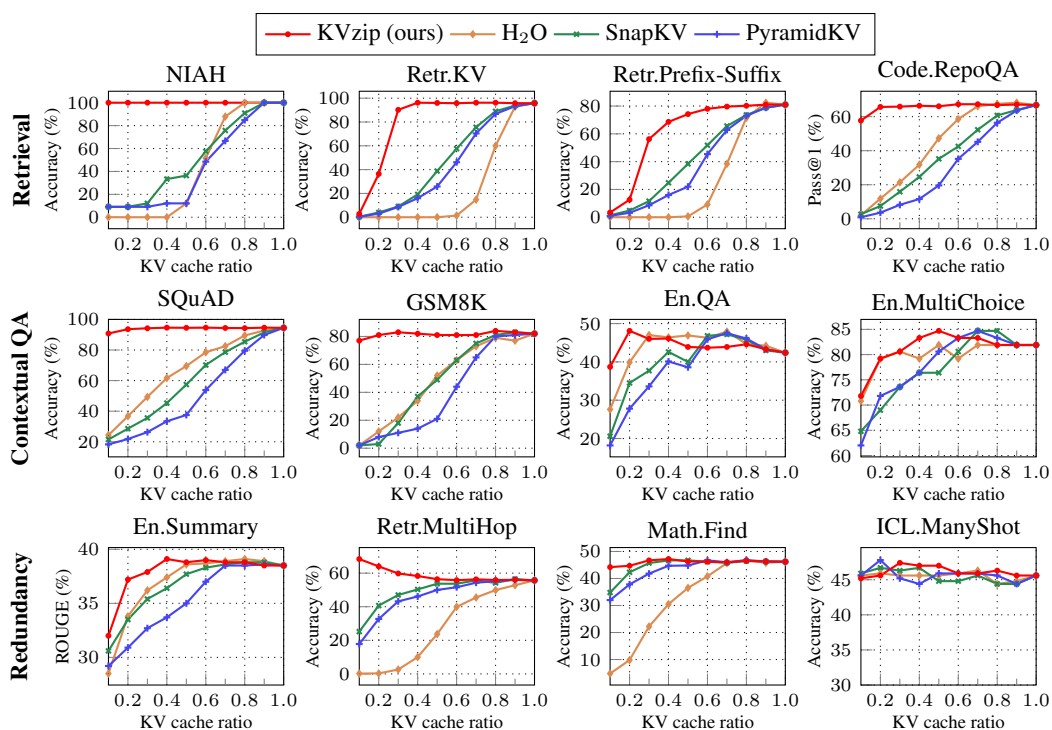

Figure 18: Benchmark results using Qwen2.5-14B-1M [54] across compression ratios from 0.1 to 1.0.

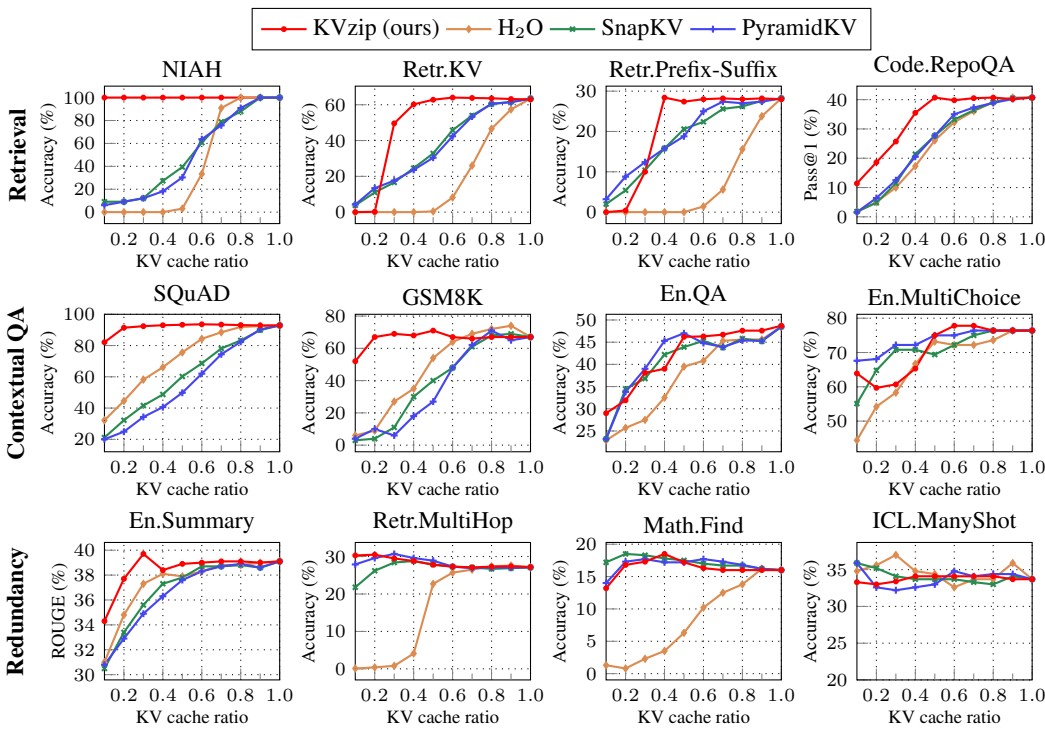

Figure 19: Benchmark results using LLaMA3.1-8B [21] across compression ratios from 0.1 to 1.0.

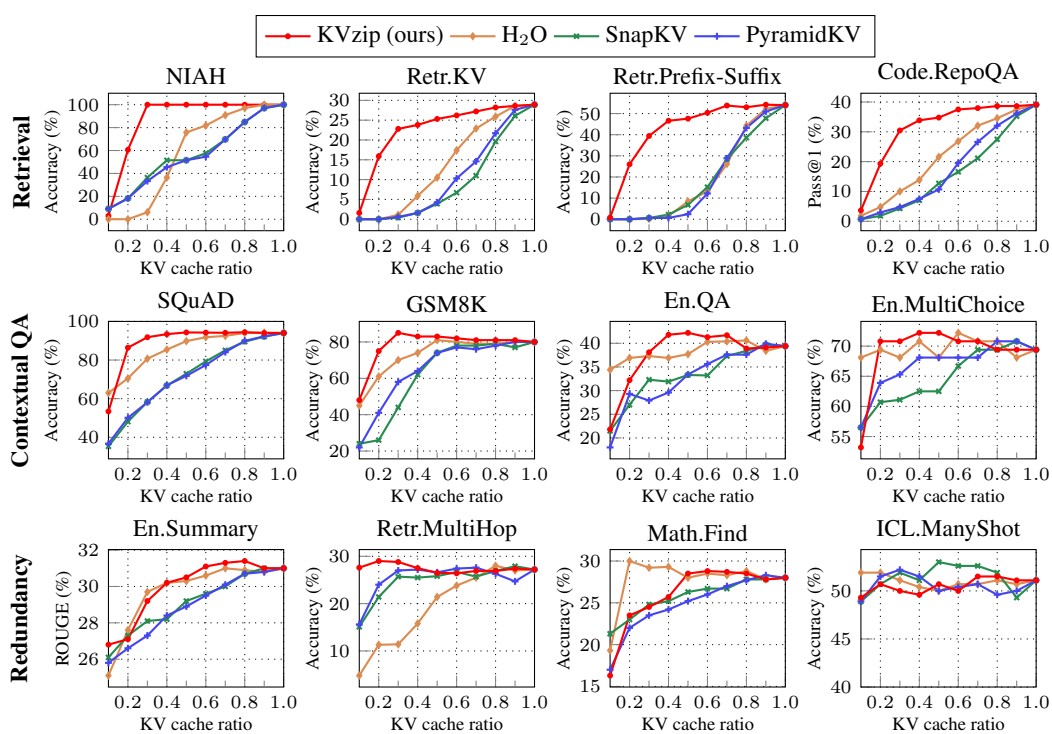

Figure 20: Benchmark results using Gemma3-12B [49] across compression ratios from 0.1 to 1.0.

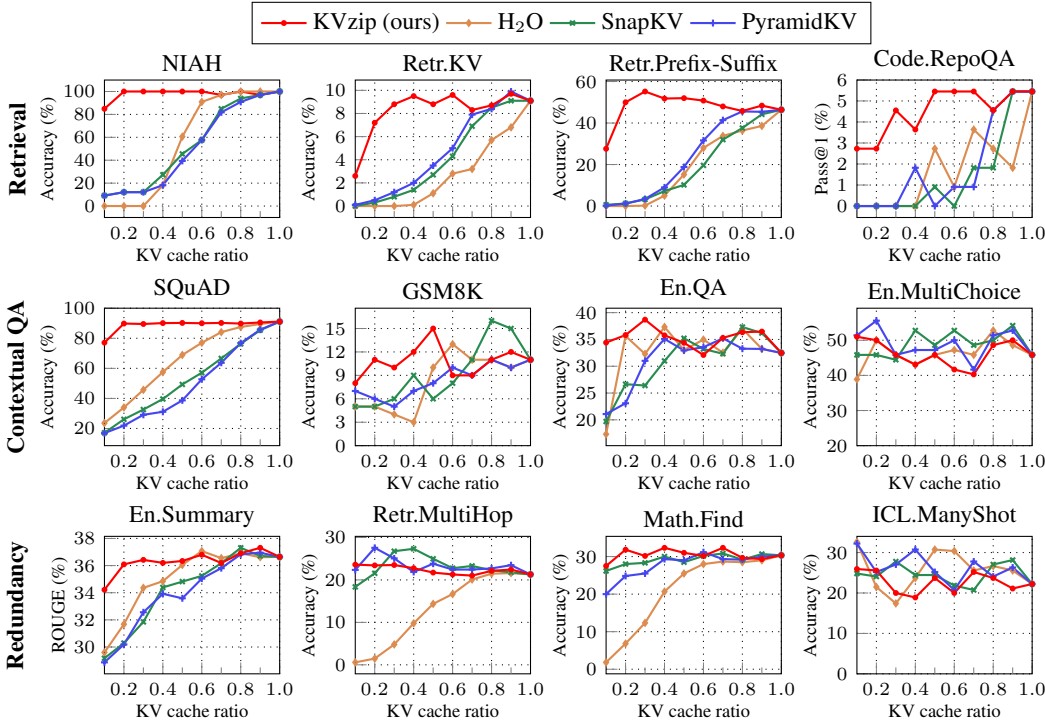

Figure 21: Benchmark results using LLaMA3-8B-W8A8KV4 [36] across compression ratios from 0.1 to 1.0.

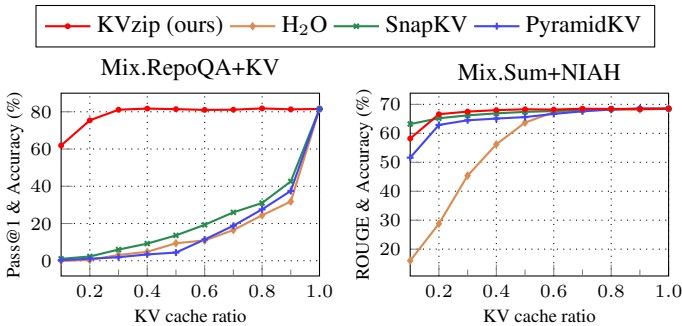

Figure 22: Benchmark results on SCBench multi-task datasets using Qwen2.5-7B-1M [54] across compression ratios from 0.1 to 1.0.

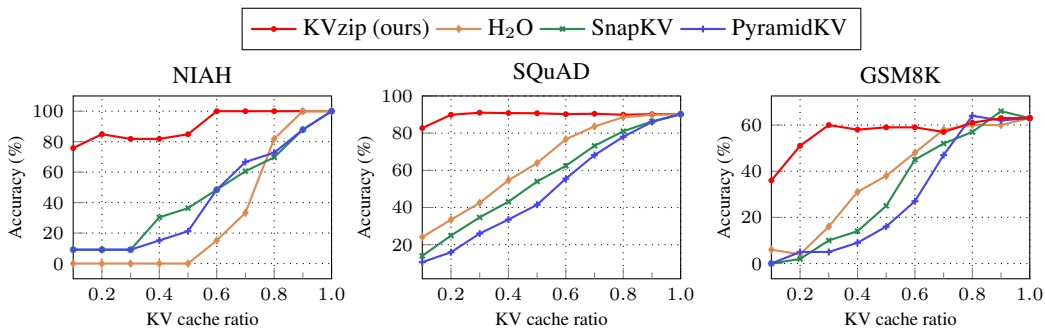

Figure 23: Benchmark results for LLaMA3.1-3B [21] across compression ratios ranging from 0.1 to 1.0. The evaluation focuses on shorter contexts, as LLaMA3.1-3B lacks the capability to solve SCBench tasks, resulting in near-zero accuracy.

Table 3: Inputs for KV cache importance scoring from a SQuAD example (used in the visualizations in Figure 6 and Figure 13). The context is included in the input of the repeat task.

| Task | Inputs for scoring (prompt+response) |
| --- | --- |
| Repeat | Repeat the previous context: Architecturally, the school has a Catholic character. Atop the Main Buildings gold dome is a golden statue of the Virgin Mary. Immediately in front of the Main Building and facing it, is a copper statue of Christ with arms upraised with the legend "Venite Ad Me Omnes". Next to the Main Building is the Basilica of the Sacred Heart. Immediately behind the basilica is the Grotto, a Marian place of prayer and reflection. It is a replica of the grotto at Lourdes, France where the Virgin Mary reputedly appeared to Saint Bernadette Soubirous in 1858. At the end of the main drive (and in a direct line that connects through 3 statues and the Gold Dome), is a simple, modern stone statue of Mary. |
| QA-1 | Q: To whom did the Virgin Mary allegedly appear in 1858 in Lourdes France? A: Saint Bernadette Soubirous |
| QA-2 | Q: What is in front of the Notre Dame Main Building? A: a copper statue of Christ |
| Summarization | Please summarize the previous context: The school has a Catholic character with various statues and buildings that reflect this theme, including a gold dome with a Virgin Mary statue, a Christ statue, a Basilica of the Sacred Heart, a Grotto replica of Lourdes, and a modern stone Mary statue at the end of the main drive. |

Table 4: Inputs for importance scoring from a GSM8K example used in the visualization in Figure 6, a reasoning task. The context is included in the input of the repeat task.

| Task | Inputs for scoring (prompt+response) |
| --- | --- |
| Repeat | Repeat the previous context: Janet's ducks lay 16 eggs per day. She eats three for breakfast every morning and bakes muffins for her friends every day with four. She sells the remainder at the farmers' market daily for $2 per fresh duck egg. |
| Reasoning | Reason and answer the question. Q: How much in dollars does she make every day at the farmers' market? Janet's ducks lay 16 eggs per day. She eats 3 eggs for breakfast, so she has 16 - 3 = 13 eggs left. She bakes 4 eggs for muffins, so she has 13 - 4 = 9 eggs left. She sells the remaining 9 eggs at the farmers' market for $2 each. To find out how much she makes, we multiply the number of eggs she sells (9) by the price per egg ($2): $9 x $2 = $18. The answer is $18. |

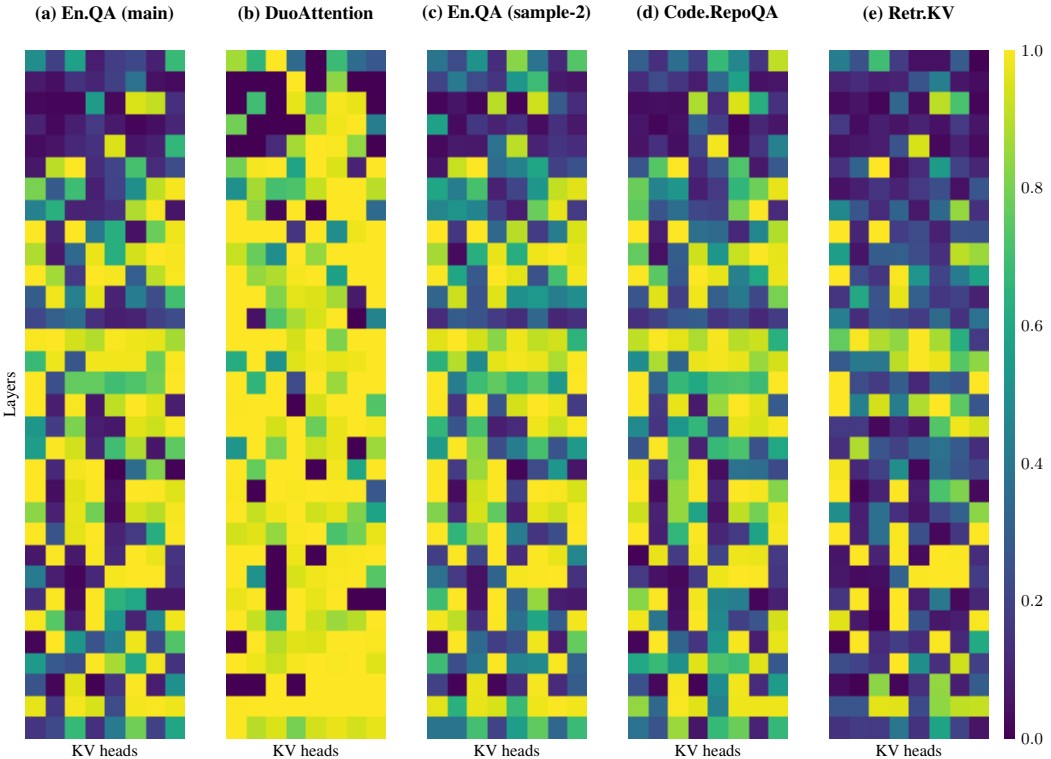

Figure 24: **Visualization of head-level importance scores** for context-independent compression in Section 4.2. We use the head scores obtained from an En.QA example in our primary experiments (Figure 11). For reference, (c)-(e) show head scores derived from alternative data sources from SCBench [35]. Our scoring method yields a more uniformly distributed importance pattern compared to DuoAttention. We select the En.QA sample for our main experiments due to its comprehensive overlap with importance patterns from other data sources, whereas Retr.KV, composed of synthetic passkeys, exhibits sparser importance patterns.

