# OpenReview forum: "KVzip: Query-Agnostic KV Cache Compression with Context Reconstruction"
_NeurIPS.cc/2025/Conference — NeurIPS 2025 oral_

### Official Review · Reviewer_64er · 2025-06-30

**Clarity:** 4
**Significance:** 4
**Originality:** 4
**Rating:** 5
**Confidence:** 4

**Summary:**

This paper introduces KVZip, a query-agnostic KV cache compression method designed for practical deployment scenarios. KVZip leverages an autoencoding concept to compress KV cache robustly without requiring knowledge of future queries. The approach is based on the observation of sparse attention patterns that emerge during the reconstruction process of a given context. Since the method is query-independent, it eliminates the need for re-prefilling when handling diverse queries on the same context, demonstrating robust and superior performance compared to existing state-of-the-art methods across various tasks and compression ratios.

**Questions:**

- *Detail* Could you provide a more detailed explanation of the intuition in Section 3.1, particularly lines 104-106 and 107-111? I would like to understand this core concept more clearly and deeply.
- *Detail* In Section 4.2's comparison with DuoAttention, could you specify exactly how head-level scores were measured and computed?
- *Related to Weakness 1* When evaluating with LLaMA-3.1's 128K context length, did positional out-of-distribution issues during the reconstruction process cause any problems? How was this addressed? (what am I missing?)
- *Suggestion* Have you compared using sum/average aggregation methods instead of max values for attention scores? Would this significantly impact performance? Additionally, is the repeat prompt robust to alternative prompt designs? A simple ablation study with basic aggregation methods and variation of repeat prompts would strengthen the paper.
- *Suggestion* The following recent works related to query-agnostic approaches might be valuable additions to the related work section;
   - Corallo et al, Beyond RAG: Task-Aware KV Cache Compression for Comprehensive Knowledge Reasoning
   - Kim et al, InfiniPot: Infinite Context Processing on Memory-Constrained LLMs

**Ethical Concerns:**

["NO or VERY MINOR ethics concerns only"]

**Final Justification:**

The authors have addressed all of my questions and concerns, and upon reviewing the other reviewers' opinions, I find that my rationale for acceptance aligns with aspects shared by other reviewers. Therefore, I will maintain my current acceptance.

**Limitations:**

Yes

**Quality:**

4

**Strengths And Weaknesses:**

**Strengths**

- The paper is well-written with clear presentation, making it easy to follow the methodology and contributions.
- The application of autoencoding concepts to KV cache compression appears highly novel, yielding robust performance results that consistently outperform existing baselines. The experimental evaluation is comprehensive, covering diverse tasks and baselines, which convincingly demonstrates the method's advantages.
- The discussion of privacy implications in KV cache compression provides a novel perspective that offers valuable insights to the research community.

Overall, the paper demonstrates high completeness with an innovative core idea and thorough reporting of both performance and efficiency aspects. This work appears to make a meaningful contribution to addressing the KV cache bottleneck in practical LLM deployment.

**Weakness**
- If my understanding is correct, the autoencoding-based scoring approach may face limitations with positional encoding when processing very long contexts. For instance, if LLaMA-3.1 supports up to 128K tokens, compressing a 128K context would require repeated input during reconstruction, potentially causing positional out-of-distribution issues. This raises concerns about whether the model can effectively utilize positional indices beyond its training length (requiring position IDs up to twice the target length for computing n_c / n_c + n_i attention maps).
- As acknowledged by the authors, KVZip is inherently prefill-heavy due to its autoencoding nature, creating somewhat significant compression overhead. While chunked prefill can reduce complexity, the method still generates KV cache footprint or lifespan [1][2] equivalent to original length plus repeated teacher-forced context. This characteristic may limit its applicability in resource-constrained on-device environments where prefill resources are highly restricted [3].

[1] Bhaskar et al, Cache me if you can: how many kva do you need for effective long-context lms?, 2506.17121
[2] Li et al, SCBench: A KV Cache-Centric Analysis of Long-Context Methods, ICLR25
[3] Kim et al, InfiniPot: Infinite Context Processing on Memory-Constrained LLMs, EMNLP 24

---

> ### Author Rebuttal · Authors · 2025-07-30
>
> Dear reviewer, thank you for your valuable time and effort in reviewing our paper. We have addressed the questions and feedback raised below. We will carefully incorporate the following discussion into our revision.
>
> ---
> **Q) Weakness 1. positional out-of-distribution issues during the reconstruction process**
> - By employing chunking in our context reconstruction method (Figure 7), we **limit the increase in position ID** to the size of each chunk (i.e., 2K tokens) during importance scoring, thus eliminating the mentioned concerns.
> - Specifically, for each chunked reconstruction, the position ID starts from the last position ID of the prefilled tokens. Since the chunk size (2K tokens) is **less than 2%** of the 128K token context length limit for LLaMA-3.1, its impact is negligible and does not introduce OOD issues for our benchmark datasets.
> - Additionally, the impact on position ID can be further reduced by decreasing the chunk size without incurring degradation in compression performance.
>
> ---
>
> **Q) KVZip is inherently prefill-heavy due to its autoencoding nature, which may limit its applicability in resource-constrained on-device environments.**
> - We emphasize that KVzip unlocks new application scenarios beyond the conventional online KV eviction setting. Specifically, KVzip produces reusable, accurately compressed KV caches suitable for multiple future queries, thus supporting **offline context KV cache compression**. This capability is particularly beneficial in real-world use cases such as enterprise document retrieval–augmented systems or personalized user contexts (lines 34–37), where the **compression overhead can be amortized** over subsequent queries. Existing dynamic KV eviction methods, however, incur significant performance degradation in multi-query scenarios, limiting their applicability in such scenarios (Figure 9).
> - Moreover, even in online settings, KVzip's importance-scoring method seamlessly extends to head-level eviction, introducing **no additional compression overhead** during deployment (see Context-Independent Eviction, Section 4.2). Compared to DuoAttention, a state-of-the-art head-level eviction technique, KVzip provides superior accuracy with substantially lower optimization costs (lines 270–275 and Figure 11).
>
> ---
>
> **Q) Could you provide a more detailed explanation of the intuition in Section 3.1, particularly lines 104-106 and 107-111?**
> - The main insight behind KVzip is ensuring that the LLM's knowledge system (KV cache + model weights) retains the **complete context information**. We believe this completeness is a necessary condition in a query-agnostic setting, as arbitrary queries may require any detail in the context. Our proposed approach validates this completeness via **prompting**.
> - [lines 104-106] Suppose we reconstruct the original context from the compressed KV cache through prompting. Successfully reconstructing the context implies that we can **regenerate** the original KV cache by performing a separate prefilling using this reconstructed context. This means we can obtain the original response (without compression) to any arbitrary query by using the compressed KV cache. This forms the intuition behind our context reconstruction-based KV cache compression method.
> - [lines 107-111] However, this approach would require a re-prefilling step, which is practically infeasible. Fortunately, we observe empirically that performing direct inference with the compressed KV cache through context reconstruction (Figure 4, right) effectively solves downstream tasks without the need for re-prefilling of the full KV cache.
> - These are detailed explanations of the requested lines. If you need any further clarification, please let us know.
>
> ---
> **Q) In Section 4.2's comparison with DuoAttention, could you specify exactly how head-level scores were measured and computed?**
> - We first obtain a KV importance score $S$ of shape $L\times H\times n_c$, where $L$ is the number of layers, $H$ is the number of heads, and $n_c$ is the context token length. We compute this score using **a single** English book sample containing 88K tokens from En.QA in SCBench (line 261). Next, we take the **maximum value along the sequence dimension** as $\max_i S[:,:,i]$, which serves as the head-level score in our experiments (line 260). The pseudocode in Appendix B describes this computation.
> - Note that we use the head-level score derived from a single sample of En.QA for all benchmark datasets, ensuring a fair comparison with DuoAttention.
>
> ---
> **Q) Have you compared using sum/average aggregation methods instead of max values for attention scores?**
> - We've tested average aggregation and compared against our max aggregation approach. Note that sum and average aggregation yield identical eviction priorities, making them equivalent.
> | Method \ Compression ratio | 0.1  | 0.2  | 0.3  | 0.5  | 1.0  |
> |----------------------------|------|------|------|------|------|
> | Max                        | 64.3 | 93.4 | 94.3 | 93.6 | 93.1 |
> | Average                    | 41.2 | 79.5 | 90.1 | 93.2 | 93.1 |
>
>   *Table. Average accuracy (%) of Qwen2.5-7B on SQuAD dataset.*
> - The results shown in the table above clearly demonstrate that max aggregation outperforms average aggregation. This empirical finding is the primary reason we adopted max aggregation in our algorithm.
> - We attribute this result to the sparsity of attention patterns. As illustrated in Figure 16 (Appendix D), LLMs tend to exhibit sparse attention distributions. Consequently, average aggregation can **dilute the significance** of tokens receiving high attention from specific queries, reducing the perceived importance of KV pairs. In analyzing cases where average aggregation failed, we found that the method often struggled with accurately retrieving words or phrases, negatively affecting various downstream LLM tasks such as QA, coding, passkey retrieval, and reasoning.
>
> ---
>
> **Q) Is the repeat prompt robust to alternative prompt designs?**
>
> - The choice of repeat prompt was motivated by simplicity, as the specific wording of the repeat prompt has **minimal impact** on overall performance.
> - To validate this, we conducted experiments comparing: (1) the original repeat prompt, (2) a paraphrased version ("Reproduce the preceding context without any changes."), and (3) no repeat prompt (just "\n\n"). Note that we follow the chat template provided by each model (e.g., "\<user> [context] [repeat prompt] \<assistant> [repeated context]").
> |Repeat prompt type|Accuracy (%)|
> |---|---|
> |Original|94.37|
> |Paraphrased|94.45|
> |No|94.25|
>
>   *Table. Accuracy (%) of Qwen2.5-7B on SQuAD at a 30% compression rate (70% eviction). SnapKV achieves **32.15%** in this setting.*
>
> - These results show that our method is **robust** to variations in the repeat prompt. The limited impact arises because the repeat prompt (7 tokens with Llama 3.1) is significantly shorter than the overall context (at least several hundred tokens), thereby minimizing its effect on compression.
> - To further clarify this, we analyzed **attention patterns**. Specifically, we measured the proportion of prefilled KV pairs whose maximum cross-attention scores during reconstruction originated from the repeated context rather than the repeat prompt (see Figure 4).
>   - For a 2K token-length context from NIAH, 98.1% of KV pairs had their maximum attention from the repeated context. Among the KV pairs retained after 30% compression, **99.4%** derived their maximum attention from the repeated context. These findings confirm the minimal influence of the repeat prompt on KVzip’s importance scores and compression performance.
> - It is also interesting to observe that context reconstruction without a repeat prompt achieves comparable performance. This indicates that forwarding the context twice effectively captures the critical attention patterns required for downstream tasks (see Figure 6).
>
> ---
> **Q) The following recent works related to query-agnostic approaches might be valuable additions to the related work section.**
> - Thank you for the suggestion! We added the suggested related works in our revision.

---

> > ### Comment · Reviewer_64er · 2025-08-03
> >
> > Thank you for the detailed response. After reading all the other reviews and rebuttals, it seems that no further concerns remain. I agree that KVzip focuses on offline context compression, and this characteristic somewhat justifies the prefill-heavy nature of this methodology. The phenomenon where the maximum method significantly outperforms averaging in score aggregation, along with the related explanation of attention sparse distribution, appears quite interesting. I believe it would be beneficial if all the discussions from this rebuttal and the additional analyses are included in the revised version. I will maintain my current score.

---

### Official Review · Reviewer_bWgi · 2025-07-01

**Clarity:** 4
**Significance:** 3
**Originality:** 4
**Rating:** 5
**Confidence:** 4

**Summary:**

Unlike recent works that rely on queries to assess the importance of Key-Value pairs, this paper introduces KVzip, a technique for quantifying the importance of KV pairs, then evicting the less important ones to effectively compress the context with minimal information losses. This method enables LLMs to perform a single prefill-eviction process, after which they can handle multi-queries without additional prefilling stage.
KVZip is especially beneficial to tackle the challenges of KV cache compression in multi-query scenarios.

**Questions:**

Please refer to the questions in the weaknesses part.

**Ethical Concerns:**

["NO or VERY MINOR ethics concerns only"]

**Final Justification:**

Few changes are made to my review. My concerns are well addressed by the authors in the rebuttal session. I would like to keep my intention for accepting this paper, as it is a technologically solid paper with innovation.

**Limitations:**

Yes

**Paper Formatting Concerns:**

No.

**Quality:**

4

**Strengths And Weaknesses:**

## Strengths
1.	This paper introduces an innovative and effective method to probe the actual importance of KV pairs. Prior works only utilizes the attention scores during the prefilling stage, but this paper utilizes the “repeat generation” task to actually see what the model needs to reconstruct the whole context, and experiments have shown superior results.

2.	The formulation, writings and figures of this paper is clear. I can clearly understand what each figure is showing immediately when I read the caption of the figure.

3.	This paper has a thorough investigation of recent works (H2O, FastGen, SnapKV, PyramidKV and DuoAtention), giving it a solid technical background.

4.	The experiments involve a wide range of benchmarks, ranging from retrieval-intensive tasks, contextual QA to tasks with high context redundancy. The evaluated model includes Qwen2.5-7B, Qwen2.5-14B, LLaMA3.1-8B, and Gemma3-12B. This broad evaluation gives it a solid support to claim the effectiveness of KVZip.

## Weaknesses
1.	The choice of the reconstruction prompt—“Repeat the previous context:”—is insufficiently justified and appears overly heuristic. This paper doesn’t explore or compare alternative formulations of the reconstruction prompt.

---

> ### Author Rebuttal · Authors · 2025-07-30
>
> Dear reviewer, thank you for your valuable time and effort in reviewing our paper. We have addressed the questions and feedback raised below. We will carefully incorporate the following discussion into our revision.
>
> ---
> **Q) The choice of the reconstruction prompt—“Repeat the previous context:”—is insufficiently justified and appears overly heuristic. This paper doesn’t explore or compare alternative formulations of the reconstruction prompt.**
>
> - This choice of repeat prompt was motivated by simplicity, as the specific wording of the repeat prompt has **minimal impact** on overall performance.
> - To validate this, we conducted experiments comparing: (1) the original repeat prompt, (2) a paraphrased version ("Reproduce the preceding context without any changes."), and (3) no repeat prompt (just "\n\n"). Note that we follow the chat template provided by each model (e.g., "\<user> [context] [repeat prompt] \<assistant> [repeated context]").
> |Repeat prompt type|Accuracy (%)|
> |---|---|
> |Original|94.37|
> |Paraphrased|94.45|
> |No|94.25|
>
>   *Table. Accuracy (%) of Qwen2.5-7B on SQuAD at a 30% compression rate (70% eviction). SnapKV achieves **32.15%** in this setting.*
>
> - These results show that our method is **robust** to variations in the repeat prompt. The limited impact arises because the repeat prompt (7 tokens with Llama 3.1) is significantly shorter than the overall context (at least several hundred tokens), thereby minimizing its effect on compression.
> - To further clarify this, we analyzed **attention patterns**. Specifically, we measured the proportion of prefilled KV pairs whose maximum cross-attention scores during reconstruction originated from the repeated context rather than the repeat prompt (see Figure 4).
>   - For a 2K token-length context from NIAH, 98.1% of KV pairs had their maximum attention from the repeated context. Among the KV pairs retained after 30% compression, **99.4%** derived their maximum attention from the repeated context. These findings confirm the minimal influence of the repeat prompt on KVzip’s importance scores and compression performance.
> - It is also interesting to observe that context reconstruction without a repeat prompt achieves comparable performance. This indicates that forwarding the context twice effectively captures the critical attention patterns required for downstream tasks (see Figure 6).

---

> > ### Comment · Reviewer_bWgi · 2025-08-05
> >
> > Thank you for your detailed response with regard to my questions and concerns. All my concerns are solved. It is indeed interesting that the performance remains strong even without the repeating prompt. This paper is novel in that it employs context reconstructing in a  parameter-free style. I would like to keep my positive score.

---

### Official Review · Reviewer_edTh · 2025-07-01

**Clarity:** 3
**Significance:** 3
**Originality:** 3
**Rating:** 5
**Confidence:** 4

**Summary:**

This paper proposes a Key-Value eviction method called KVzip. The approach preserves the most informative key-value vectors by ensuring that the compressed key-value cache enables the LLM to reconstruct the original text context.

**Questions:**

Please refer to Weaknesses.

**Ethical Concerns:**

["NO or VERY MINOR ethics concerns only"]

**Final Justification:**

I have read the response and raised my score to 5. The authors provide more experimental results, which can resolve my concerns.

**Limitations:**

Yes

**Quality:**

3

**Strengths And Weaknesses:**

[Strengths]
1. Intuitive and well-motivated approach: The method's core intuition is reasonable and straightforward. The experimental evaluation demonstrates competitive performance across numerous downstream tasks, validating the effectiveness of the proposed approach.

2. Comprehensive experimental validation: The authors provide extensive experiments to demonstrate the method's effectiveness, including evaluations across multiple models, datasets, and compression ratios. The empirical analysis is thorough and convincing.

[Weaknesses]
1. Significant computational overhead: The key-value eviction process itself is computationally expensive and requires substantial time overhead. As shown in Figure 8, the compression overhead is more than 3 times the original prefilling time. Therefore, for long document inputs, this method would only be cost-effective when there are at least 3 or more queries per document to amortize the compression cost.

2. Limited evaluation scope: The authors' evaluation focuses primarily on scenarios with one document, same task type, and multiple different queries. However, the paper lacks evaluation of scenarios involving one document with multiple queries from different task types, which would provide a more comprehensive assessment of the method's generalizability across diverse query patterns.

---

> ### Author Rebuttal · Authors · 2025-07-30
>
> Dear reviewer, thank you for your valuable time and effort in reviewing our paper. We have addressed the questions and feedback raised below. We will carefully incorporate the discussion into our revision.
>
> ---
>
> **Q) The key-value eviction process itself is computationally expensive and requires substantial time overhead.**
> - We emphasize that KVzip unlocks new application scenarios beyond the conventional online KV eviction setting. Specifically, KVzip produces reusable, accurately compressed KV caches suitable for multiple future queries, thus supporting **offline context KV cache compression**. This capability is particularly beneficial in real-world use cases such as enterprise document retrieval–augmented systems or personalized user contexts (lines 34–37), where the **compression overhead can be amortized** over subsequent queries. Existing dynamic KV eviction methods, however, incur significant performance degradation in multi-query scenarios, limiting their applicability in such scenarios (Figure 9).
> - Moreover, even in online settings, KVzip's importance-scoring method seamlessly extends to head-level eviction, introducing **no additional compression overhead** during deployment (see Context-Independent Eviction, Section 4.2). Compared to DuoAttention, a state-of-the-art head-level eviction technique, KVzip provides superior accuracy with substantially lower optimization costs (lines 270–275 and Figure 11).
> ---
>
> **Q) The authors' evaluation focuses primarily on scenarios with one document, same task type, and multiple different queries. However, the paper lacks evaluation of scenarios involving one document with multiple queries from different task types.**
> - Thank you for the pointer. In the Appendix, Figure 22 (described in text in lines 555-558), we present experimental results for the **multi-task, multi-query setting**. The evaluation uses two multi-task benchmarks from SCBench: Mix.RepoQA+KV and Mix.Sum+NIAH [1]. The following highlighted results demonstrate that KVzip consistently outperforms the baselines in a multi-task setting:
> | task            | H2O  | SnapKV | PyramidKV | KVzip | Full KV cache |
> |-----------------|------|--------|-----------|-------|---------------|
> | Mix.RepoQA+KV   | 3.1  | 6.0    | 1.9       | 81.1  | 81.5          |
> | Mix.Sum+NIAH    | 45.4 | 66.2   | 64.5      | 67.5  | 68.5          |
>
>   *Table. Multi-task, multi-query performance (%) at a 30% compression ratio using Qwen2.5-7B.*
>
> - Notably, KVzip is not only query-agnostic but also **task-agnostic**, as the compression procedure does not depend on task types. This implies that our main experiment in Section 4 using single-task benchmarks spanning 12 different tasks effectively demonstrates the task-generalizability of KVzip, which is why we placed the additional results on multi-task benchmarks in the Appendix. We will highlight these results in the main text in our revision.
>
> [1] Li et al., SCBench: A kv cache-centric analysis of long-context methods. ICLR, 2025.

---

### Official Review · Reviewer_yysm · 2025-07-02

**Clarity:** 3
**Significance:** 2
**Originality:** 2
**Rating:** 5
**Confidence:** 5

**Summary:**

KVzip proposes a query-agnostic eviction scheme for long-context LLMs that uses a prompt of the form [context + ‘Repeat the previous context’ + context], and then uses (max-pooled) attention scores of the second copy of the context (on the first copy of the context) to generate ‘importance scores’ for each KV pair. The top-r% highest scoring KV pairs are retained. The aforementioned procedure is computationally expensive, so the authors propose chunking the second copy of the context and using multiple passes through the model to generate the scores. They empirically show that this scoring procedure outperforms baseline methods (PyramidKV, SnapKV, H2O) on a variety of benchmark tasks. They show that using a smaller KV cache (with any eviction method) provides non-trivial latency improvements (section 3.4). They show that KVzip produces a significant increase in prefilling latency (section 3.4), and demonstrate the entire second context copy is needed for KVzip to provide good performance (Section 4.3). Lastly, they show that utilizing KVzip can modify the behavior tuning of pre-trained models with a privacy-leaking example.

**Questions:**

What inference platform/engine/package was used to determine the numbers in Figure 8?

Can you provide additional “Computational analysis” (Figure 8) for larger models (20B+)? Running all the benchmark evaluations isn’t necessary, but it would be nice to know how the performance scales.

Do you have any additional insights into how the latency of the method can be improved (would a fused kernel help significantly here)?

How was the exact form of the prompt settled on and what alternatives were tried? Normally this wouldn’t be important, but given the reliance of the model on this particular prompt, it would be nice to know.

The NIAH you selected is quite easy (it is an obviously incongruous phrase inserted into a random essay); how does KVzip perform on a more difficult task   (multikey, multivalue, uuid retrieval from RULER)?

**Ethical Concerns:**

["NO or VERY MINOR ethics concerns only"]

**Final Justification:**

My response is largely unchanged but the authors have done their best to respond to those concerns. If the paper were to be accepted, their additional materials here would bolster the presentation of the work, which I continue to find on the edge of practicality.

**Limitations:**

Yes

**Quality:**

3

**Strengths And Weaknesses:**

Strengths

Query-agnostic compression is an important scenario that has good practical importance

The proposed solution is simple, and would be easy to reproduce

The paper is clearly written, the method is well explained, and there is a good variety of experiments to support their claims. The figures
and captions are clear and well done (Figure 6 especially).

The section on context-independent eviction obviates the need for expensive procedures for generating head-level eviction policies

Weaknesses

The solution is not especially novel; it is essentially an application of SnapKV to a particular prompt structure, with speed improvements provided by chunking. The text-reconstruction/auto-encoder approach is also not new, though I haven’t seen this exact approach in the parameter-free setting before.

For practical applications, the prefill latency increase induced by KVzip is incredibly high, and is probably untenable in many scenarios. Also, the number of forward passes through the model (while conditioning on the whole–supposedly long–context that needs to be compressed) scales linearly as context length grows, especially with high parameter count models. This is a major limitation of the KVzip.

---

> ### Author Rebuttal · Authors · 2025-07-30
>
> Dear reviewer, thank you for your valuable time and effort in reviewing our paper. We have addressed the questions and feedback raised below. We will carefully incorporate the discussion into our revision.
>
> ---
>
> **Q) The solution is essentially an application of SnapKV to a particular prompt structure.**
> - We note that our method introduces a query-independent objective, **proactively identifying** crucial KV pairs for arbitrary future queries. This represents a clear methodological distinction from prior works, such as SnapKV and H2O, which compress based solely on the current query, leading to **query-specific overfitting**. This distinction significantly enhances practical utility: KVzip maintains inference accuracy at a 30% compression rate, whereas prior methods exhibit severe performance degradation in query-agnostic scenarios (e.g., a 60% accuracy drop on the SQuAD dataset with Qwen2.5-7B at the same compression rate, see Figure 9).
> ---
>
> **Q) The reconstruction approach is also not new, though I haven’t seen this exact approach in the parameter-free setting before.**
> - While reconstruction-based objectives are well-established in representation learning, we emphasize that connecting this principle explicitly to parameter-free KV compression is novel, as acknowledged by the reviewer. This **novel connection** significantly enhances practical applicability in query-agnostic scenarios, enabling near-lossless compression at a 30% rate. In contrast, prior methods incur notable performance degradation even at a much higher 90% compression rate (see Figure 9).
> ---
>
> **Q) The prefill latency increase induced by KVzip is incredibly high, and is probably untenable in many scenarios.**
> - We emphasize that KVzip unlocks new application scenarios beyond the conventional online KV eviction setting. Specifically, KVzip produces reusable, accurately compressed KV caches suitable for multiple future queries, thus supporting **offline context KV cache compression**. This capability is particularly beneficial in real-world use cases such as enterprise document retrieval–augmented systems or personalized user contexts (lines 34–37), where the **compression overhead can be amortized** over subsequent queries. Existing dynamic KV eviction methods, however, incur significant performance degradation in multi-query scenarios, limiting their applicability in such scenarios (Figure 9).
> - Moreover, even in online settings, KVzip's importance-scoring method seamlessly extends to head-level eviction, introducing **no additional compression overhead** during deployment (see Context-Independent Eviction, Section 4.2). Compared to DuoAttention, a state-of-the-art head-level eviction technique, KVzip provides superior accuracy with substantially lower optimization costs (lines 270–275 and Figure 11).
> ---
>
> **Q) What inference platform/engine/package was used to determine the numbers in Figure 8?**
> - We used PyTorch with Hugging Face Transformers and FlashAttention2 on an NVIDIA A100 GPU. This setup follows the convention used in previous works such as SnapKV, PyramidKV, and KVpress.
> ---
>
> **Q) Can you provide additional “Computational analysis” (Figure 8) for larger models (20B+)?**
> - Following your suggestion, we conducted computational analysis using **Qwen3-32B (FP8)** with a context length of 124K tokens on an NVIDIA H100 GPU. (Note: Qwen3’s FP8 quantization does not support the A100 GPU used previously.) We employed identical software packages as used in Figure 8.
> - The table below demonstrates that KVzip maintains similar **efficiency gains** with the larger model, achieving approximately a 2$\times$ speed-up in decoding attention latency at a 20% compression rate.
> | Compression ratio | Decoding attention latency (ms) | KV cache size (GB) |
> |---|---|---|
> |1|0.45|32.7|
> |0.8|0.39|26.2|
> |0.6|0.34|19.6|
> |0.4|0.28|13.1|
> |0.2|0.23|6.5|
>
>   *Table. Computational efficiency analysis of KVzip during decoding.*
>
> - Regarding **compression overhead**, KVzip introduces approximately 1.5$\times$ computational overhead, slightly lower than the 2$\times$ overhead observed with the 8B model in Figure 8. Specifically, KVzip’s chunked context reconstruction process requires 1$\times$ MLP FLOPs and 2$\times$ attention FLOPs relative to prefilling. As model size increases, the relative computational contribution of MLP layers grows, thereby reducing the relative overhead of the scoring process.
> - Memory overhead from KVzip’s scoring process **remains negligible** compared to prefilling (0.8% increase in peak memory), consistent with results observed on smaller models.
> |Stage|Compute time (s)|Peak GPU memory (GB)|
> |---|---|---|
> |prefill|3m 56s|62.7|
> |scoring (compression)|5m 25s|63.2|
>
>   *Table. Computational comparison of prefilling and KV importance scoring stages. We use a chunk size of 2K for both stages to ensure a fair comparison.*
> ---
>
> **Q) Do you have any additional insights into how the latency of the method can be improved (would a fused kernel help significantly here)?**
> - We explored various strategies to reduce compression overhead. While a tailored fused CUDA kernel could help reduce latency, the benefit is relatively modest (~10%, line 528 in Appendix) since the primary overhead arises from the FLOPs by the Transformer forward-pass on repeated contexts.
> - One such solution in our paper to further reduce latency is **context-independent eviction** (Line 257), which eliminates additional latency at deployment.
> - We also tested the method that **interleaves** chunked prefilling and compression (i.e., compress the KV cache of the i-th chunk before prefilling the next chunk) to reduce overall attention FLOPs during prefilling. However, this led to decreased compression performance. In our paper, we prioritize achieving optimal KV-cache compression quality and present head-level context-independent eviction as a practical alternative. Exploring further improvements in prefilling latency remains part of our future work.
> ---
>
> **Q) How was the exact form of the prompt settled on and what alternatives were tried?**
> - In our code, we used the repeat prompt: "\n\nRepeat the previous context exactly." This choice was motivated by simplicity, as the specific wording of the repeat prompt has **minimal impact** on overall performance.
> - To validate this, we conducted experiments comparing: (1) the original repeat prompt, (2) a paraphrased version ("Reproduce the preceding context without any changes."), and (3) no repeat prompt (just "\n\n"). Note that we follow the chat template provided by each model (e.g., "\<user> [context] [repeat prompt] \<assistant> [repeated context]").
> |Repeat prompt type|Accuracy (%)|
> |---|---|
> |Original|94.37|
> |Paraphrased|94.45|
> |No|94.25|
>
>   *Table. Accuracy (%) of Qwen2.5-7B on SQuAD at a 30% compression rate (70% eviction). SnapKV achieves **32.15%** in this setting.*
>
> - These results show that our method is **robust** to variations in the repeat prompt. The limited impact arises because the repeat prompt (7 tokens with Llama 3.1) is significantly shorter than the overall context (at least several hundred tokens), thereby minimizing its effect on compression.
> - To further clarify this, we analyzed **attention patterns**. Specifically, we measured the proportion of prefilled KV pairs whose maximum cross-attention scores during reconstruction originated from the repeated context rather than the repeat prompt (see Figure 4).
>   - For a 2K token-length context from NIAH, 98.1% of KV pairs had their maximum attention from the repeated context. Among the KV pairs retained after 30% compression, **99.4%** derived their maximum attention from the repeated context. These findings confirm the minimal influence of the repeat prompt on KVzip’s importance scores and compression performance.
> - It is also interesting to observe that context reconstruction without a repeat prompt achieves comparable performance. This indicates that forwarding the context twice effectively captures the critical attention patterns required for downstream tasks (see Figure 6).
> ---
>
> **Q) The NIAH you selected is quite easy; how does KVzip perform on a more difficult task (multikey, multivalue, uuid retrieval from RULER)?**
> - Following your recommendation, we evaluated KVzip on **RULER**. Specifically, we integrated KVzip into NVIDIA’s KVpress framework, which benchmarks state-of-the-art KV eviction methods under a query-agnostic setting with a 4K context length RULER using LLaMA3.1-8B-Instruct.
> - The table below compares KVzip to SnapKV on the suggested challenging tasks from RULER. Results indicate that KVzip maintains **perfect performance** at a 25% compression rate, whereas SnapKV experiences significant performance degradation.
> |task|SnapKV|KVzip|Full KV cache|
> |---|---|---|---|
> |Multi key-uuid|1.6|100.0|100.0|
> |Multi value|25.3|99.9|99.9|
>
>   *Table. RULER-4K performance (%) at a 25% compression ratio.*
>
> - To further highlight KVzip's effectiveness, we compared our method with various benchmark results from KVpress, which include **state-of-the-art methods** such as SnapKV, PyramidKV, DuoAttention, and other **concurrent unpublished methods**. While these results are publicly available on Hugging Face, the link is excluded here due to rebuttal policy constraints. The following table demonstrates that KVzip significantly outperforms current state-of-the-art KV eviction methods:
> |Compression ratio|SnapKV|Best from KVpress|KVzip|
> |---|---|---|---|
> |0.5|67.0|91.6|95.5|
> |0.25|39.9|74.3|95.3|
>
>   *Table. RULER-4K average performance (%) over 13 tasks. The full cache performance is **95.5**%.*
>
> - Lastly, our primary experiments in Figure 9 demonstrate KVzip's capability to handle significantly **longer contexts** (up to 170K tokens) on SCBench, which also includes representative tasks from RULER.

---

### Decision · Program_Chairs · 2025-09-17

**Decision:**

Accept (oral)

**Comment:**

All reviewers highlighted the paper’s practical motivation, clear presentation, and comprehensive experiments. The method was recognized as simple, reproducible, and innovative in applying context reconstruction for query-agnostic KV cache compression, with strong results across tasks. Concerns centered on heavy prefill latency and computational overhead, possible positional encoding issues at very long contexts, limited initial coverage of multi-task settings, and the heuristic nature of the reconstruction prompt. In the rebuttal, the authors provided additional analyses, including larger-model experiments, multi-task evaluations, and robustness tests on RULER and prompt variations, which addressed most of these weaknesses. Overall, the reviewers agreed the empirical validation was thorough and the methodological contribution clear. Given its novelty, practical relevance, and broad interest for the community, the recommendation is acceptance.